# Cis- and trans-resveratrol have opposite effects on histone serine-ADP-ribosylation and tyrosine induced neurodegeneration

Megha Jhanji [1], Chintada Nageswara Rao [1], Jacob C. Massey [1], Marion C. Hope III[1], Xueyan Zhou[2], C. Dirk Keene [3], Tao Ma [2], Michael D. Wyatt [1], Jason A. Stewart[4] & Mathew Sajish [1] ✉

Serum tyrosine levels increase during aging, neurocognitive, metabolic, and cardiovascular disorders. However, calorie restriction (CR) and sleep lower serum tyrosine levels. We previously showed that tyrosine inhibits tyrosyl-tRNA synthetase (TyrRS)-mediated activation of poly-ADP-ribose polymerase 1 (PARP1). Here, we show that histone serine-ADP-ribosylation is decreased in Alzheimer's Disease (AD) brains, and increased tyrosine levels deplete TyrRS and cause neuronal DNA damage. However, dopamine and brain-derived neurotrophic factor (BDNF) increase TyrRS and histone serine-ADP-ribosylation. Furthermore, cis-resveratrol (cis-RSV) that binds to TyrRS mimicking a 'tyrosine-free' conformation increases TyrRS, facilitates histone serine-ADP-ribosylation-dependent DNA repair, and provides neuroprotection in a TyrRS-dependent manner. Conversely, trans-RSV that binds to TyrRS mimicking a 'tyrosine-like' conformation decreases TyrRS, inhibits serine-ADP-ribosylation-dependent DNA repair, and induces neurodegeneration in rat cortical neurons. Our findings suggest that age-associated increase in serum tyrosine levels may effect neurocognitive and metabolic disorders and offer a plausible explanation for divergent results obtained in clinical trials using resveratrol.

[1] Department of Drug Discovery and Biomedical Sciences, College of Pharmacy, University of South Carolina, Columbia, SC 29208, USA. [2] Department of Internal Medicine, Gerontology and Geriatric Medicine, Wake Forest School of Medicine, Winston-Salem, NC, USA. [3] Department of Laboratory Medicine and Pathology, University of Washington School of Medicine, Seattle, WA, USA. [4] Department of Biological Sciences, College of Arts and Sciences, University of South Carolina, Columbia, SC 29208, USA. ✉email: mathew2@cop.sc.edu

Tyrosyl-tRNA synthetase (TyrRS, also known as YARS1) belongs to the family of aminoacyl-tRNA synthetases (aaRSs) and activates the aromatic amino acid (AAA) tyrosine for protein synthesis[1]. Although tyrosine exists as enantiomers (L- and D- tyrosine) and TyrRS can activate both[2], only L-tyrosine is utilized for protein synthesis[1,2]. Similarly, only L-Tyr (Tyr), not D-Tyr, is a substrate for the rate-limiting step in dopamine synthesis[3] catalyzed by tyrosine hydroxylase (TH). Serum tyrosine is circadian-regulated, with the highest levels in the morning and lowest at midnight (sleep time)[4]. Intriguingly, brain protein synthesis[5], memory formation[6], and neuronal DNA repair[7] are activated during sleep when tyrosine levels are decreased during the nadir/trough of the circadian rhythm. Tyrosine levels are also modulated by the circadian activities of TH, tyrosine aminotransferases (TAT), and gut microbiota. Genetic mutations that increase tyrosine levels (tyrosinemia) or its precursor L-phenylalanine (Phe, phenylketonuria [PKU]) cause multiple health problems, including cognitive deficits in children. Moreover, tyrosine exacerbates the cognitive decline in elderly[8] and Alzheimer's disease (AD) patients[9] and drives axonal degeneration and demyelination in tyrosinemia patients[10]. Although protein synthesis is required for long-term memory formation[11] and brain-derived neurotrophic factor (BDNF) stimulates the de novo synthesis of TyrRS in neurons[12], recent brain proteomic analysis[13] showed that TyrRS is decreased in the affected brain regions of AD patients[13] through an unknown mechanism. Therefore, identification of endogenous factors and their mechanisms of action that modulate TyrRS levels may provide deeper insights into the pathophysiology of AD and other neurocognitive and metabolic disorders.

Calorie restriction (CR) promotes genomic stability through the induction of base excision repair (BER) and reversal of its age-related decline[14] along with an extension of lifespan and protection against age-associated neurocognitive and metabolic disorders, including cardiovascular diseases (CVD). Although metabolic analyses show that CR decreases serum tyrosine levels[15] and tyrosinemia patients have shortened lifespan[10], whether tyrosine directly modulates aging and age-associated disorders remains unknown. Most importantly, the natural molecule resveratrol (RSV) that shows similarity to tyrosine (both contain phenolic group) demonstrated CR-like health benefits in humans[16], suggesting that RSV may act as a 'CR mimetic'[16]. Intriguingly, clinical studies using the trans-isomer of RSV (trans-RSV) revealed conflicting outcomes. Lower doses of trans-RSV produced encouraging results, but higher doses exacerbated the diseases. For example, low-dose trans-RSV showed CR-like benefits in obese males[16], cognitive benefits in AD patients[17] and postmenopausal women[18], protection against heart failure[19], and cancer chemoprevention[20]. However, higher doses (≥200 mg/day) of trans-RSV resulted in brain volume loss in AD patients[21], worsened memory performance in schizophrenia[22], worsened the metabolic profile in diabetic patients[23], and increased CVD risk in older adults[24]. Despite decades of research, the molecular basis of these controversial effects of trans-RSV (low dose CR-like beneficial effects[16–20] versus high dose detrimental effects[21–24]) remains unknown[25]. Although RSV exists as a mixture of cis-RSV and trans-RSV, recent studies showed that the sulfate metabolites of trans-RSV that provide an intracellular pool[26] mainly generate cis-RSV[27]. We previously showed that cis- and trans-RSV mimic tyrosine and bind to TyrRS[28]. However, cis-RSV induces a unique 'tyrosine-free' conformation in TyrRS[25]. Since CR decreases tyrosine levels[15], we proposed that cis-RSV may enable the moonlighting nuclear functions of TyrRS even in the presence of tyrosine and therefore, cis-RSV may act as a potential 'CR mimetic'[25,28].

We previously showed that tyrosine inhibits TyrRS-mediated auto-poly-ADP-ribos(PAR)ylation of poly-ADP-ribose polymerase 1 (PARP1) and associated stress signaling[28]. Consistently, auto-PARylation of PARP1 is circadian-regulated in a feeding-dependent manner, in which feeding that increases tyrosine levels inhibits auto-PARylation[29]. These observations suggested that the nuclear functions of TyrRS are typically activated when tyrosine level is decreased during the nadir/trough of the circadian rhythm. However, we also showed that lower concentration trans-RSV adapts its cis conformation (cis-RSV) to activate TyrRS-dependent auto-PARylation of PARP1[28]. Although auto-PARylation of PARP1 is essential for BER[30] and sleep activates neuronal DNA repair[7], whether cis-RSV would enable TyrRS/PARP1-dependent DNA repair is not yet known. Moreover, the apparent Ki value of trans-RSV-mediated inhibition of tyrosine activation by TyrRS in an ATP-PPi exchange assay (Tyr + ATP → Tyr-AMP + PPi) is ~25 μM[28], suggesting that trans-RSV may retain its trans conformation (mimicking 'tyrosine-like' conformation) at higher concentrations (≥25 μM)[25,28]. However, whether trans-RSV would inhibit TyrRS/PARP1-dependent DNA repair, especially the auto-PARylation of PARP1[25,28], is not yet known. Most importantly, clinical studies using 5 and 1000 mg of trans-RSV (>99% trans-RSV) reported peak plasma concentrations of 0.6 and 137 μM of RSV respectively[20], and other clinical studies using 1000 mg/day of trans-RSV for 29 days reported an accumulation of 50–640 μM of trans-RSV in human tissues[26], suggesting that this treatment regimen could achieve high dose (≥25 μM) trans-RSV-mediated effects in humans.

We found that increased tyrosine levels decrease TyrRS and cause neuronal oxidative DNA damage by simultaneously inhibiting protein synthesis and DNA repair. However, dopamine and BDNF stimulated the de novo synthesis of neuronal TyrRS. Furthermore, we show that cis- and trans-RSV have opposite effects on TyrRS levels, protein synthesis, DNA repair, and survival of rat cortical neurons. cis-RSV protects the neurons against tyrosine and other neurotoxic agents-induced depletion of TyrRS and DNA damage. Most importantly, we show that trans-RSV evokes dichotomic effects depending on dose, exerting neuroprotection at lower concentrations (≤10 μM) but inducing and exacerbating neurotoxicity at higher concentrations (≥25 μM). This dichotomic effect correlated with isomeric transitions of trans-RSV in solution and TyrRS levels, where most of trans-RSV gets converted to cis-RSV at low concentrations[27,28] that increase TyrRS levels and activate neuronal DNA repair (potentially mimicking CR-like protective effects) but remains in the trans conformation at high concentrations potentially causing tyrosine-like toxic effects such as depletion of TyrRS, inhibition of protein synthesis and induction of DNA damage-associated neurodegeneration. Therefore, beyond explaining the opposite effects of the high- and low-dose trans-RSV found in clinical trials, our results also provide a potential molecular basis for the age-associated increase in neuronal DNA damage and cognitive decline due to age-associated increase in serum tyrosine levels.

## Results

**TyrRS is decreased in the hippocampal tissue samples of human AD patients.** AD decreases brain protein synthesis at the elongation step in humans[31–34] through an unknown mechanism. Recently published human brain proteome showed decreased TyrRS and phenylalanyl-tRNA synthetase beta (PheRSβ) levels in AD-affected brain regions[13] (Supplementary Fig. 1a, b). We validated the observation that TyrRS and PheRSβ are depleted in the hippocampal region of AD patients. However, the level of PheRSα was not affected (Fig. 1a and Supplementary Table 1). Our re-analysis of a second brain proteome[35] showed that the protein levels of TyrRS and PheRSβ correlate with cognitive performance in humans (Supplementary Fig. 1c)[35]. Conversely, their decrease correlates with AD status and Braak stages

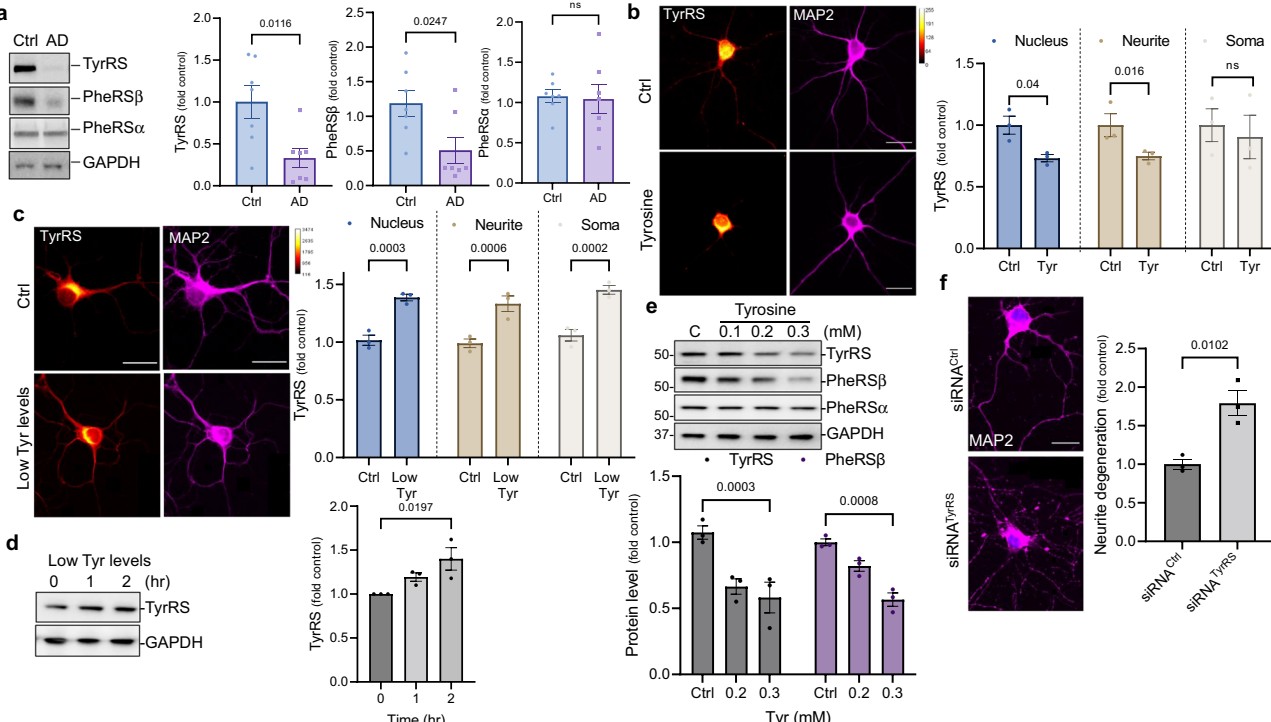

**Fig. 1 TyrRS protein is decreased in the hippocampal region of AD patients, and tyrosine decreases nuclear and neurite levels of TyrRS.**
**a** Representative immunoblots and quantification for TyrRS and PheRSα/β using anti-TyrRS and PheRSα/β antibodies in the hippocampal region of AD patients ($n = 7$) with age and sex-matched controls ($n = 7$). **b** Tyrosine preferentially decreases nuclear and neurite levels of TyrRS. Spectral images (scale bar, 20 μm) and quantitative IF analysis of TyrRS in the nucleus, soma, and neurite of rat cortical neurons (DIV 9) using anti-TyrRS antibody after treatment with L-tyrosine (250 μM) for 4 hr ($n = 3$). **c** Reduction in tyrosine level increases the nuclear and neurite levels of TyrRS. Spectral images (scale bar, 20 μm) and quantitative IF analysis of TyrRS in the nucleus, soma, and neurite of rat cortical neurons (DIV 9) using anti-TyrRS antibody after treatment with low tyrosine medium for 2 hr as described in methods ($n = 3$). **d** Reduction in tyrosine levels increase total TyrRS protein levels. Representative immunoblots showing total TyrRS after treatment with low tyrosine medium for 2 hr in rat cortical neurons (DIV 9) using anti-TyrRS antibody ($n = 3$). **e** Tyrosine depletes TyrRS and PheRSβ, but not PheRSα. Rat cortical neurons were treated with L-tyrosine (100–300 μM) for 4 hr. The levels of PheRSβ and PheRSα were detected by WB analysis using their specific antibodies ($n = 3$). **f** TyrRS knockdown using siRNA induces neurite degeneration. Representative images (scale bar, 20 μm) for cortical neurons 72 hr following siRNA$^{TyrRS}$ transfection (MAP2 – neurite marker, magenta, and DAPI – nuclear marker, blue). Neurons were immunoassayed with anti-MAP2 antibody and quantified for neurite degeneration. The data is presented as mean ± SEM for $n = 3$ experiments. Statistical significance was measured using a two-tailed unpaired t-test. Source data are provided as a Source Data file.

(Supplementary Fig. 1d)[35]. Intriguingly, a meta-analysis of human brain transcriptomic data from AD patients did not show any changes in the mRNA levels of TyrRS and PheRSβ[36]. While the translation of PheRSβ declines in an age-dependent manner (Supplementary Fig. 1e)[37], TyrRS levels did not correlate with any known biomarkers of AD or other neurodegenerative diseases (Supplementary Fig. 1f)[35], indicating that hitherto unknown factors might modulate neuronal TyrRS protein levels.

**Tyrosine is increased during aging and in neurocognitive and metabolic disorders in humans.** Aging is the strongest risk factor for neurodegenerative diseases, and intriguingly, the incidence of AD and other dementias is higher in women than in men. Serum tyrosine levels increase during aging, and tyrosine levels are also increased in AD brain tissues. Interestingly, young women have lower serum tyrosine levels than young men. However, menopause increases serum tyrosine levels, resulting in a significant increase in serum tyrosine levels in older women (Supplementary Fig. 2a and Supplementary Table 2). Beyond neurodegenerative diseases, aging in humans is also associated with increased incidences of other diseases. Consistently, our literature analysis showed that tyrosine and/or phenylalanine are increased during delirium, heart failure (HF), Parkinson's Disease (PD), autism

spectrum disorders (ASD), CVD, and other metabolic disorders (Supplementary Fig. 2b and Supplementary Table 3). Interestingly, increased tyrosine and/or phenylalanine levels inhibit protein synthesis[38,39], induce DNA damage[40], and promote oxidative stress in the brain[41] in vivo in rats. Because AD decreases TyrRS levels (Fig. 1a) and brain protein synthesis[31–34], we hypothesized that increased tyrosine levels might negatively regulate TyrRS protein levels and cause neuronal oxidative DNA damage.

**Tyrosine facilitates the degradation of neuronal TyrRS by both proteosome and lysosome in rat cortical neuronal cultures.** To determine whether treatment with tyrosine modulates TyrRS levels, we treated rat cortical neurons (DIV 9/10) with increasing tyrosine concentrations. Consistent with our hypothesis, tyrosine decreased TyrRS in neurons (DIV 9/10) (Fig. 1b) with preferential effects in the nucleus and the neurites (Fig. 1b). Conversely, reducing tyrosine levels in the culture medium increased TyrRS levels in the nucleus and neurites (Fig. 1c, d). Although tyrosine depleted PheRSβ, it did not affect PheRSα levels (Fig. 1e). Similarly, phenylalanine, 3,4-dihydroxy-L-phenylalanine (L-DOPA), and 6-hydroxydopamine (6-OHDA) also decreased TyrRS and PheRSβ levels (Supplementary Fig. 3a–c), suggesting that derivatives of tyrosine metabolism also negatively regulate

TyrRS. Albeit an essential protein, we previously showed that ~75% knockdown of TyrRS using siRNA (siRNA$^{TyrRS}$) does not affect cell viability[28]. However, siRNA$^{TyrRS}$ in cortical neurons (~50% knockdown) resulted in robust neurite degeneration (Fig. 1f and Supplementary Fig. 3d), indicating a critical role of TyrRS in maintaining neurite stability. In addition, recently published mass spectrometry data showed that TyrRS remains heavily ubiquitinated in the cell[42] and its deubiquitination (along with other aaRSs) facilitates the recovery of protein synthesis during the recovery phase after stress[42]. To determine if tyrosine exploits the proteasomal or lysosomal pathway (autophagy) for TyrRS degradation along with its ability to inhibit protein synthesis[38,39], we performed additional experiments using inhibitors of autophagy (bafilomycin), proteasome (MG132), and protein synthesis (cycloheximide, CHX). Treatment with both bafilomycin and MG132 increased TyrRS levels, suggesting that both the proteasome and lysosome are involved in the constitutive degradation of TyrRS (Supplementary Fig. 3e, f). However, cycloheximide decreased TyrRS levels (Supplementary Fig. 3g), suggesting that sustained de novo synthesis is required to maintain the homeostatic levels of TyrRS under normal conditions. In contrast, deubiquitination may stabilize TyrRS under stress conditions[42].

**Dopamine mimics BDNF in stimulating the de novo synthesis of TyrRS protein in rat cortical neuronal cultures**. Protein synthesis is mainly regulated at the initiation and elongation steps. Ser51 phosphorylation of eukaryotic initiation factor 2 alpha (p-eIF2α) by multiple kinases and Thr56 phosphorylation of eukaryotic elongation factor 2 (p-eEF2) by eEF2 kinase (eEF2K) inhibit protein synthesis at the initiation and elongation steps, respectively. The protein kinase mammalian target of rapamycin (mTOR) inhibits eEF2K and activates protein synthesis (Supplementary Fig. 4a). Although it is counter-intuitive that elevated tyrosine level is inhibitory for TyrRS and protein synthesis in a p-eIF2α-independent manner[38,39], for which it is required, we also noted that tyrosine co-instantaneously activates the assembly of eukaryotic initiation factor 4 F (eIF4F) and phosphorylation of ribosomal protein S6 kinase beta-1 (S6K1)[39]. More importantly, despite a significant decrease in the neurite TyrRS levels, surprisingly, tyrosine did not induce neurite degeneration (Fig. 1b), and TyrRS levels were restored in 16–24 h (Fig. 2a). These observations suggest that the effect of tyrosine on TyrRS is reversible and neurotrophic factors that activate protein synthesis may stimulate the de novo synthesis of TyrRS. The neurotransmitter dopamine (DA), which is decreased during aging, and in the affected brain regions of AD patients, is generated from tyrosine (L-Tyr → L-DOPA → DA). Interestingly, DA is known to activate eEF2[43], potentially by activating mTOR, stimulating both protein synthesis and memory formation[44]. Because BDNF, which is also depleted in AD brains, activates eEF2 and stimulates the de novo synthesis of TyrRS[12] (Supplementary Fig. 4b, c), we wondered if DA would also increase TyrRS levels. Consistent with our hypothesis, treatment with DA increased TyrRS levels (Fig. 2b and Supplementary Fig. 4d), and the effects of DA were abrogated by rapamycin (Rapa) (Fig. 2c). Although tyrosine does not affect eIF2α function[39], treatment with tyrosine inhibited eEF2 (Fig. 2d). These data suggest that beyond facilitating the degradation of TyrRS, tyrosine also inhibits the de novo synthesis of TyrRS, potentially at the elongation step through increased phosphorylation of eEF2, whereas BDNF and DA stimulate TyrRS synthesis (Fig. 2e).

**Cis-RSV and trans-RSV have opposite effects on TyrRS levels in rat cortical neuronal cultures**. Natural RSV exists as a mixture

of cis-RSV and trans-RSV, and both are stable for at least six weeks at 4 °C and in the cell for 24 hr[27]. Clinical studies using low-dose trans-RSV (50–75 mg/dose, i.e.; 6–9 µM plasma concentration of trans-RSV/dose) reported CR-like benefits in postmenopausal women[18] and patients with heart failure[19]. In contrast, clinical studies with 1,000 mg/dose trans-RSV resulted in brain volume loss in AD patients[21] and increased CVD risk in older adults[24]. However, we noted that a higher tyrosine level is a common biomarker for all the diseases/conditions mentioned above (Supplementary Table 3). Because brain TyrRS levels correlate with cognitive performance in humans (Supplementary Fig. 1c)[35], we hypothesized that cis-RSV and trans-RSV would exert differential effects on neuronal TyrRS levels and protein synthesis. Consistent with the results in clinical trials, a low dose of trans-RSV (10 µM) increased TyrRS, and a high dose (50 µM) decreased it, whereas cis-RSV (10–50 µM) increased TyrRS levels both in the nucleus and neurites (Fig. 3a, b and Supplementary Fig. 5a, b), mimicking the effects of reduced tyrosine level that increase TyrRS in the nucleus and neurites (Fig. 1c). cis-RSV rescued the effects of trans-RSV and tyrosine-mediated decrease in TyrRS levels in a dose-dependent manner (Fig. 3c, d). High concentration trans-RSV decreased PheRSβ, not PheRSα levels, while cis-RSV increased their levels (Fig. 3e and Supplementary Fig. 5c, d). Further, cis-RSV-mediated increase in TyrRS level was abrogated by CHX (Fig. 3f), suggesting that cis-RSV increases TyrRS in a protein synthesis-dependent manner. While cis-RSV triggered a transient increase in the levels of p-eIF2α (Supplementary Fig. 5e, f), trans-RSV sustained the inhibition of eIF2α (Supplementary Fig. 5e, f) and increased p-eEF2 levels (Fig. 3g). However, cis-RSV activated the dephosphorylation of eEF2 (Fig. 3g). Finally, consistent with the depletion of TyrRS and increased p-eIF2α and p-eEF2, treatment with trans-RSV inhibited global protein synthesis as measured by the puromycin incorporation assay (Fig. 3h). Because CR decreased tyrosine levels[15] and decreased tyrosine levels increased TyrRS, taken together, these data indicate that cis-RSV may act as a potential CR mimetic by increasing TyrRS levels.

Because eEF2 regulates the elongation phase of protein synthesis and eEF2K is the only known kinase that inhibits eEF2 (Supplementary Fig. 4a), we wondered if direct pharmacological activation of eEF2K using nelfinavir[45] would deplete TyrRS. Interestingly, nelfinavir depleted neuronal TyrRS, which was rescued by cis-RSV (Supplementary Fig. 6a, b), suggesting that protein synthesis is required to maintain TyrRS levels under normal conditions. We previously showed that genetic reduction of eEF2K attenuates age-related memory deficits in mice[46]. Consistently, we found that TyrRS and PheRSβ were increased in the brain tissue samples of eEF2K$^{+/−}$ mice (Supplementary Fig. 6c). To determine the mechanism of cis-RSV-mediated activation of eEF2 through dephosphorylation, we tested if cis- and trans-RSV modulate the interaction of TyrRS with eEF2 and protein phosphatase 2 (PP2A) (Supplementary Fig. 4a). We found that cis-RSV facilitated the interaction of eEF2 with TyrRS and PP2A, whereas trans-RSV and tyrosine decreased their interactions (Fig. 3i). These data suggest that cis-RSV and tyrosine modulate TyrRS levels at the elongation step and cis-RSV mimics DA and BDNF in activating the de novo synthesis of TyrRS (Fig. 2e). Furthermore, similar to 6-OHDA (Supplementary Fig. 3c), other neurotoxic agents such as N-Methyl-D-aspartate (NMDA), and the mitochondrial toxin 1-methyl-4-phenylpyridinium (MPP$^+$) also decreased the levels of neuronal TyrRS after 4 hr of treatment (Supplementary Fig. 6d, e); cis-RSV suppressed this effect, whereas trans-RSV exacerbated it (Supplementary Fig. 6d, e), suggesting that neuronal TyrRS is a potential target of multiple neurotoxic agents.

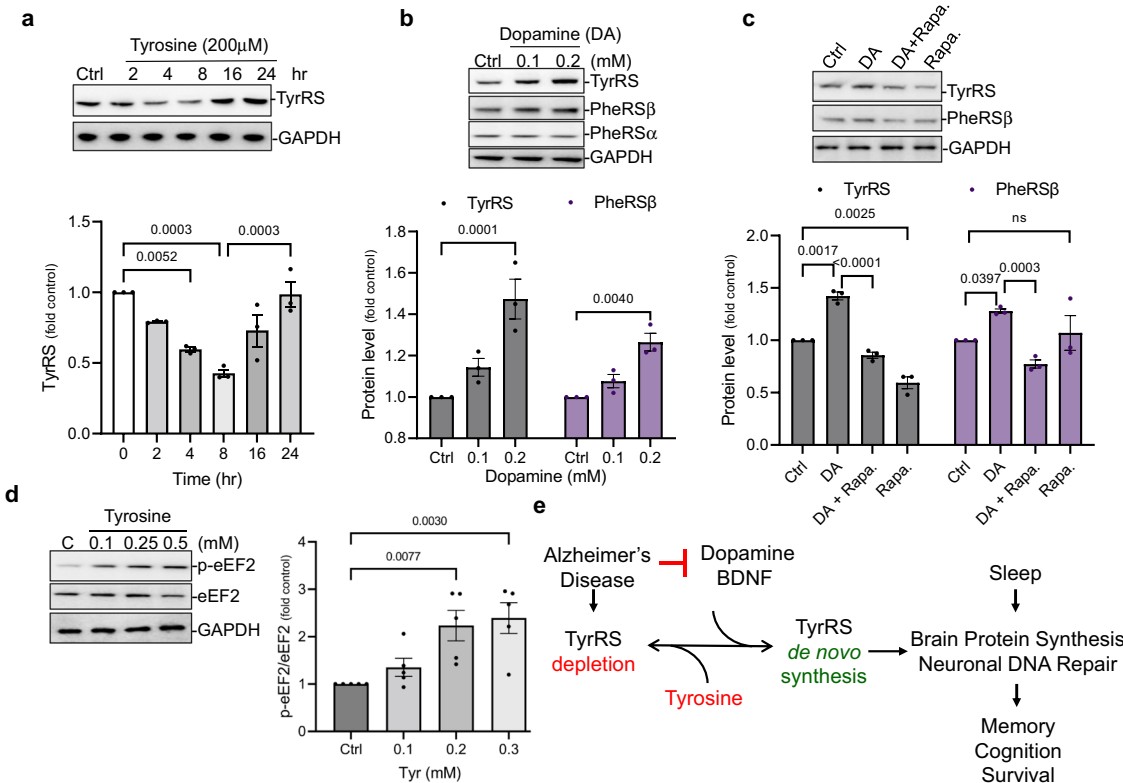

**Fig. 2 Dopamine activates the de novo synthesis of neuronal TyrRS. a** *L-Tyrosine effect on TyrRS protein levels is reversible.* Rat cortical neurons were treated with L-tyrosine (200 μM) for up to 24 hr, and TyrRS was detected by WB analysis using anti-TyrRS antibody. **b** *Dopamine (DA) increases neuronal TyrRS levels.* Representative immunoblots for TyrRS and PheRSα/β after treatment with DA (100–200 μM) for 30 min in rat cortical neurons (DIV 9). **c** *Rapamycin prevents DA-induced increase in TyrRS levels.* Representative immunoblots for TyrRS and PheRSβ after 30 min treatment with either DA (100 μM) alone or in combination with rapamycin (100 nM) in rat cortical neurons (DIV 9). **d** *Tyrosine induces the phosphorylation of eEF2.* Primary cortical neurons were treated with L-tyrosine (100–300 μM) for 8 hr and p-eEF2 was detected by WB analysis using anti-p-eEF2 antibody (n = 5). **e** *Schematic representation of the potential mechanism of regulation of the neuronal protein levels of TyrRS.* All data represent mean ± SEM for n = 3 experiments with significance measured using two-way ANOVA with Tukey's test for multiple comparisons. Source data are provided as a Source Data file.

**Pharmacological activation of protein synthesis increases TyrRS levels in rat cortical neurons.** Regulation of eIF2α modulates motor and cognitive functions, and neuronal survival in an activating transcription factor 4 (ATF4)-dependent mechanism[11]. Interestingly, TyrRS (but not PheRSβ) is among the ATF4 target genes upregulated during integrated stress response (ISR)[47]. Because pharmacological activation of protein synthesis using integrated stress response inhibitor (ISRIB) protects against age-related memory deficits[48], we tested if ISRIB would protect against *trans*-RSV-mediated depletion of TyrRS. Treatment with ISRIB (5–50 nM) increased the protein levels of TyrRS and PheRSβ (Fig. 4a), rescued *trans*-RSV-mediated depletion of TyrRS (Fig. 4b), and stimulated the dephosphorylation of eEF2 (Fig. 4a), potentially mediated through TyrRS. However, higher doses of ISRIB (250–500 nM) decreased TyrRS levels (Fig. 4c). Since we have no indication that the regulation of gene expression during ISR is different in neurons, and ISRIB inhibits ATF4 target gene expression[48], the concentration-dependent decrease in TyrRS level is likely due to the repression of TyrRS at higher doses of ISRIB, providing a potential molecular basis for ISRIB-mediated toxic effects in animal models. As tyrosine decreased neurite TyrRS levels (Fig. 1b), and sleep stimulates synaptic protein synthesis[49], we also wondered if the synaptic TyrRS is circadian-regulated. Our re-analysis of the mouse circadian proteomic and metabolomic data showed that synaptic protein level of only TyrRS (among all the aaRSs) is circadian-regulated and is inversely correlated with tyrosine levels (Supplementary Fig. 7a)[49]. Further, a re-analysis of the human metabolome

showed that sleep deprivation increases serum tyrosine levels (Supplementary Fig. 7b). Collectively, these data suggest that tyrosine is a potential endogenous modulator of the synaptic and nuclear TyrRS.

**Tyrosine induces oxidative DNA damage and *cis*-RSV protects against it in rat cortical neurons.** Human aging and neurodegenerative diseases accumulate oxidative DNA damage-associated mutations in neurons[50] whereas CR protects against oxidative stress[14]. While sleep deprivation causes oxidative DNA damage, sleep stimulates neuronal DNA repair[7]. Because TyrRS has a role in DNA damage response (DDR) signaling[28], we hypothesized that decreased tyrosine levels at night might switch the function of a fraction of TyrRS from protein synthesis to DNA repair. Conversely, aging and neurodegenerative diseases that increase tyrosine levels (Supplementary Fig. 2) might inhibit TyrRS-mediated DNA repair. Consistent with our hypothesis, treatment with tyrosine resulted in the accumulation of γ-H2AX foci (a marker of DNA damage) and 8-oxo-2'-deoxyguanosine (8-oxo-dG, a marker of oxidative DNA damage) (Fig. 5a, b). These observations indicated that compounds that mimic 'tyrosine-like' conformation in TyrRS might induce oxidative DNA damage. Conversely, compounds that mimic a 'tyrosine-free' conformation in TyrRS may shift its function to facilitate neuronal DNA repair. To test this possibility, we treated neuronal cultures with *cis*-RSV, which evokes a 'tyrosine-free' conformation[25], and *trans*-RSV, which mimics a 'tyrosine-like' conformation in TyrRS[28]. While *cis*-RSV rescued tyrosine-induced accumulation

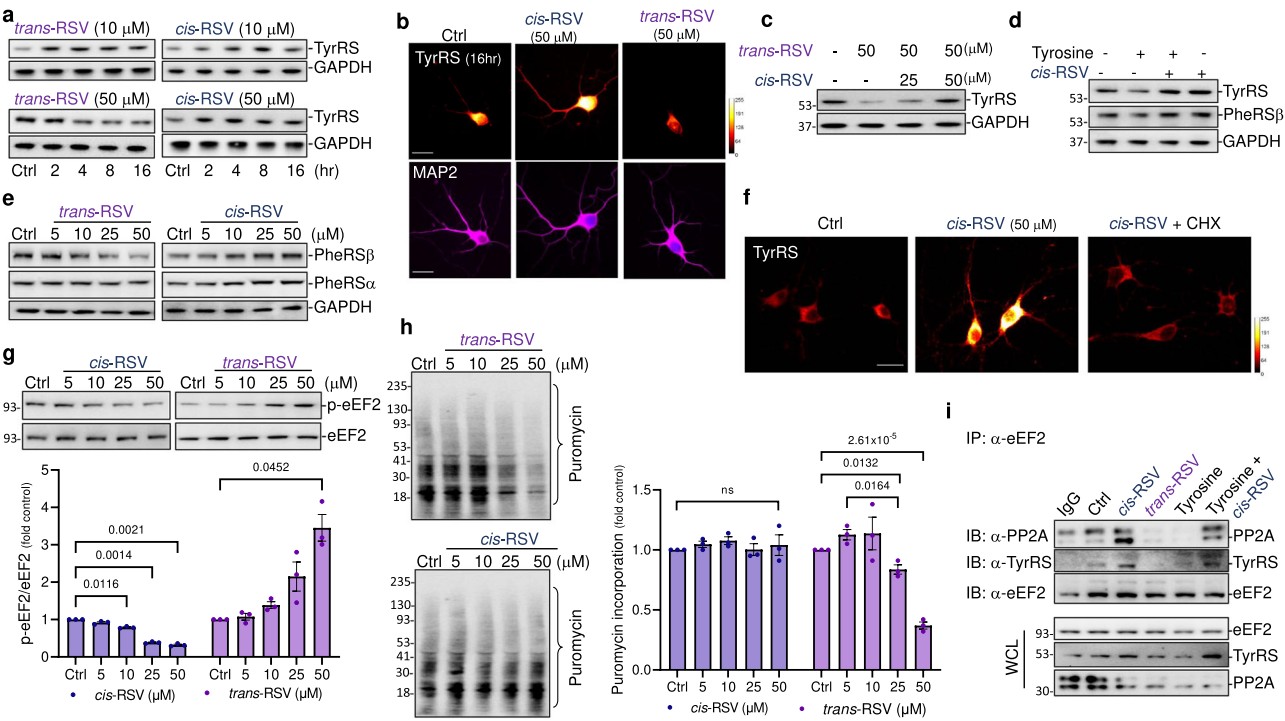

**Fig. 3 Tyrosine and *cis*-RSV modulate the de novo synthesis of TyrRS at the elongation step of protein synthesis. a** *Low and high concentrations of trans-RSV (10 and 50 μM) have opposite effects on TyrRS, and low concentration trans-RSV mimics cis-RSV.* Representative immunoblots for TyrRS after treatment with *cis*-and *trans*-RSV (10 and 50 μM) in rat cortical neurons (DIV 9). **b** *cis*-and *trans*-RSV have opposite effects on TyrRS. Representative spectral images (scale bar, 20 μm) of TyrRS in rat cortical neurons (DIV 10) following treatment with either *cis*-RSV or *trans*-RSV for 16 hr (MAP2 – neurite, magenta; DAPI – nucleus, blue; TyrRS – red-yellow spectral image). **c** *cis*-RSV protects neurons against trans-RSV-mediated decrease in TyrRS. Representative immunoblots showing TyrRS after treatment with *trans*-RSV (50 μM) alone or combined with different doses of *cis*-RSV for 4 hr. **d** *cis-RSV protects neurons against tyrosine-mediated depletion of TyrRS.* Cortical neurons were treated with Tyr (200 μM) alone or combined with *cis*-RSV for 8 hr. **e** *cis- and trans-RSV have opposite effects on neuronal PheRSβ.* Representative immunoblots for PheRSα/β after treatment with *cis*-and *trans*-RSV (5–50 μM) for 4 hr. **f** *cis-RSV stimulates the* de novo *synthesis of TyrRS.* Spectral images (scale bar, 20 μm) after treatment with *cis*-RSV (50 μM) for 1 hr either alone or combined with CHX (100 μg). **g** *cis- and trans-RSV have opposite effects on the phosphorylation of eEF2.* Representative immunoblots for p-eEF2 after treating with *cis*- and *trans*-RSV (50 μM) for 8 hr. **h** *trans-RSV inhibits global protein synthesis.* Cortical neurons were treated with *cis*-and *trans*-RSV (5–50 μM) for 1 hr, followed by puromycin incorporation and detection using anti-puromycin antibody. **i** *cis-RSV facilitates the interaction of PP2A with eEF2.* Cortical neurons were treated with *cis*- and *trans*-RSV (50 μM) or Tyr (200 μM) either alone or combined with *cis*-RSV for 4 hr. Immunoprecipitated eEF2 was probed for its interaction with PP2A and TyrRS. All data represent mean ± SEM for *n* = 3 experiments with significance measured using two-way ANOVA with Tukey's test for multiple comparisons. Source data are provided as a Source Data file.

of γ-H2AX, and 8-oxo-dG, *trans*-RSV itself caused the accumulation of γ-H2AX and 8-oxo-dG (Fig. 5a, b). Interestingly, *cis*-RSV reduced γ-H2AX levels in neuronal cultures even after 12 hr of tyrosine pre-treatment that had already caused substantial DNA damage (Supplementary Fig. 8a), suggesting that beyond prevention, *cis*-RSV may reverse existing neuronal DNA damage. Further, tyrosine decreased 8-oxoguanine-DNA glycosylase (OGG1) levels (Supplementary Fig. 8b), which was rescued by *cis*-RSV (Supplementary Fig. 8c). 8-oxo-dG is highly mutagenic, driving a G·C → T·A transversion. Consistently, mutagenic frequency increases during aging and γ-H2AX and 8-oxo-dG accumulate in aged and AD neurons[50,51]. Further, D-Tyr, which does not get converted to DA[3] but is activated by TyrRS[2], induced neurotoxicity, whereas D-Phe and D-Trp had no significant effects (Fig. 5c and Supplementary Fig. 8d, e). *cis*-RSV protected against D-Tyr-mediated neurotoxicity, whereas *trans*-RSV exacerbated the toxicity (Supplementary Fig. 8f). We also conducted the comet assay to gain direct evidence for DNA damage. We treated rat cortical neurons with *trans*-RSV and L-tyrosine either alone or in combination. We found that both *trans*-RSV, L-tyrosine, and their combination significantly increased DNA damage as measured by the increase in percentage (%) of DNA in the comet tail (Fig. 5d). In contrast, *cis*-RSV treatment did not cause

an increase in percentage of DNA in the comet tail, and attenuated the increase in DNA damage caused by Tyr (Fig. 5d). Since CR activates BER[14], taken together these data indicate that *cis*-RSV may act as a *CR mimetic* by activating BER.

**Histone serine-ADP-ribosylation is decreased in the hippocampal tissue samples of human AD patients and BDNF and dopamine increase its levels in rat cortical neurons.** Histone poly-ADP-ribosylation factor (HPF1)-dependent serine-ADP-ribosylation is essential for PARP1-dependent DNA repair and histone H3 is one of the best-characterized targets of HPF1/PARP1-mediated serine-ADP-ribosylation[52]. Because AD brains show accumulation of neuronal DNA damage[51], we wondered if HPF1 and histone serine-ADP-ribosylation levels are affected in AD brains. Our analysis showed reduced HPF1 levels along with decreased histone H3 serine-ADP-ribosylation in the hippocampal tissues of AD patients (Fig. 6a). Because HPF1 mRNA level is not affected in AD brains (Supplementary Fig. 8g) and tyrosine inhibits protein synthesis[38,39], next we tested if treatment with tyrosine and *trans*-RSV would modulate HPF1 at the translational level. Similarly, we also determined if treatment with DA, BDNF or ISRIB would affect HPF1 levels. We found that

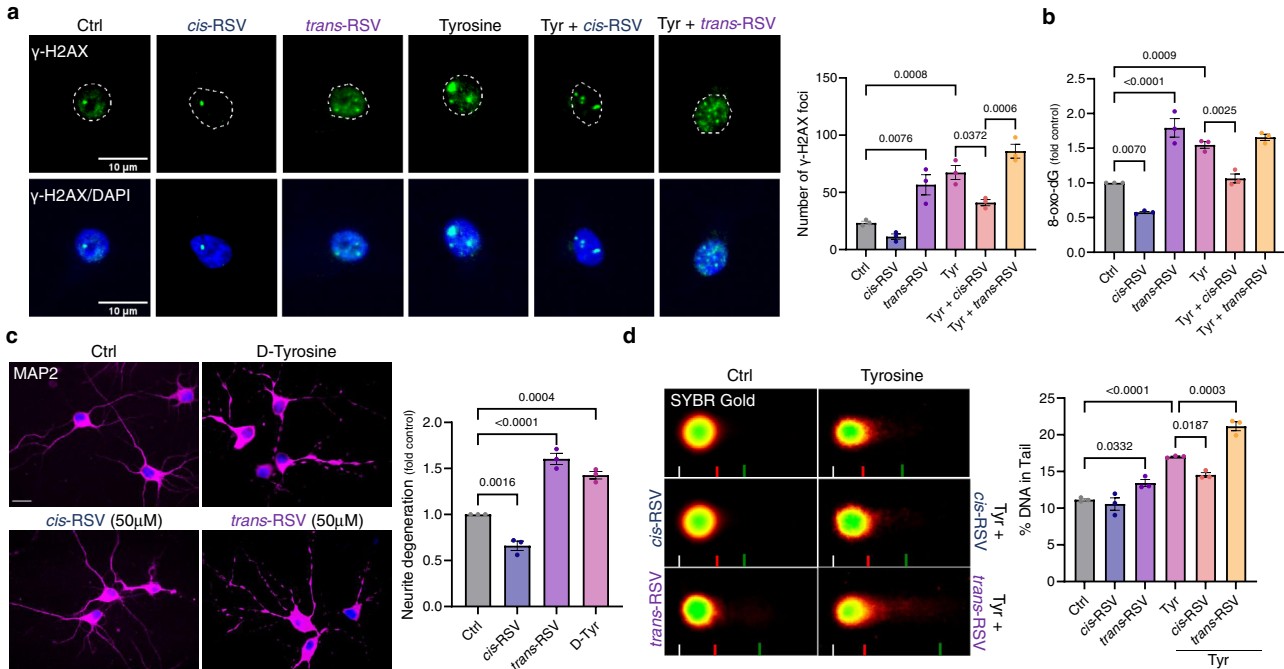

**Fig. 4 Tyrosine induces oxidative DNA damage in neurons, and *cis*-RSV protects neurons against it. a** *Cis-RSV and trans-RSV have opposite effects on tyrosine-mediated accumulation of γ-H2AX*. Immunostaining images (scale bar, 10 μm) for DNA damage marker, pSer139-H2AX foci (γ-H2AX, green; DAPI – nuclear marker, blue) in cortical neurons (DIV 10) after treatment with *cis*- and *trans*-RSV (50 μM) alone or in combination with L-tyrosine (250 μM) for 24 hr. The graph represents the average number of γ-H2AX foci per $n = 30$ neurons per treatment condition for $n = 3$ experiments. **b** *cis*- and *trans*-RSV have opposite effects on tyrosine-mediated induction of the oxidative damage in the neuronal DNA. Quantification of the levels of 8-oxo-2′-dG using immunofluorescence (IF) in rat primary cortical neurons (DIV 9/10) after treatment with L-tyrosine (500 μM) either alone or in combination with *cis* or *trans*-RSV (50 μM) for 16 hr for $n = 3$ experiments. **c** *D-tyrosine and trans-RSV induce neurite degeneration*. Representative images (scale bar, 20 μm) for cortical neurons following D-tyrosine or *cis*-RSV and *trans*-RSV (50 μM) for 24 hr treatment (MAP2 – neurite marker, magenta and DAPI – nuclear marker, blue). Neurons were immunoassayed with anti-MAP2 antibody and quantified for neurite degeneration. **d** *Comet assay measuring the extent of DNA strand breaks induced by tyrosine and trans-RSV*. Cortical neurons (DIV 9/10) were treated with *cis*-RSV (50 μM), *trans*-RSV (50 μM), L-tyrosine (1 mM) either alone or in combination for 1 hr, and the percentage of DNA in the comet tails was quantified as described in the Methods. All data represent mean ± SEM for $n = 3$ independent experiments with significance measured using two-way ANOVA with Tukey's test for multiple comparisons.

while treatment with *cis*-RSV stimulated serine-ADP-ribosylation, treatment with tyrosine and *trans*-RSV decreased HPF1 levels along with the inhibition of histone H3 serine-ADP-ribosylation (Fig. 6b–e). Furthermore, treatment with either BDNF or ISRIB increased HPF1 levels along with the induction of histone H3 serine-ADP-ribosylation (Fig. 6f, g). Similarly, DA also stimulated histone H3 serine-ADP-ribosylation (Fig. 6h). Taken together, beyond DNA repair, our data indicate a potential role of serine-ADP-ribosylation in cognition and memory and provide a potential molecular basis for the reduced HPF1 levels and serine-ADP-ribosylation in the hippocampal tissues of AD patients.

***Cis*-RSV is neuroprotective and *trans*-RSV is neurotoxic in rat cortical neuronal cultures**. After showing that *cis*- and *trans*-RSV have opposite effects on neuronal oxidative DNA damage, we hypothesized that they would exert differential effects on neuronal survival under stress conditions. We analyzed the effects of *cis*- and *trans*-RSV on the survival of rat primary cortical neurons exposed to different stress agents to test this hypothesis. As expected, the effect of *trans*-RSV on NMDA-mediated neurotoxicity showed a concentration-dependent dual response, where low concentrations of *trans*-RSV (≤10 μM) evoked protective effects, but the higher concentrations (≥25 μM) exacerbated the toxicity (Fig. 7a). In contrast, *cis*-RSV protected against NMDA-mediated toxicity in a concentration-dependent manner (Fig. 7b). Hence, the concentration-dependent dual response of *trans*-RSV

on neuroprotection is consistent with *trans* to *cis* conversion at low concentrations[27,28], increasing TyrRS levels (Fig. 3a), and the retention of 'trans/tyrosine-like' conformation at high concentrations[25,28], causing TyrRS depletion (Fig. 3a). We also found that *cis*-RSV (50 μM) suppressed the neurotoxicity induced by a DNA-damaging agent (etoposide, ETO) (Supplementary Fig. 9a), oxidative stress ($H_2O_2$) (Supplementary Fig. 9b), and mitochondrial inhibition ($MPP^+$) (Supplementary Fig. 9c) but *trans*-RSV (50 μM) did not protect against these neurotoxic agents (Supplementary Fig. 9a–c). We further confirmed that *cis*-RSV-mediated rescue of ETO-mediated DNA damage is reflected by decreased levels of γ-H2AX (Supplementary Fig. 9a). To test if the observed effects of *cis*- and *trans*-RSV on neurotoxicity are mediated via TyrRS, we carried out siRNA knockdown of TyrRS ($siRNA^{TyrRS}$) in rat cortical neurons. Although TyrRS knockdown (Supplementary Fig. 4d) did not significantly affect the viability, it blunted the neuroprotective effects of *cis*-RSV and did not diminish the toxicity of *trans*-RSV (50 μM) upon NMDA treatment (Fig. 7c). These results suggest that the neuroprotective effect of *cis*-RSV (and low-dose *trans*-RSV) is TyrRS dependent, but the neurotoxic effect of *trans*-RSV is TyrRS independent. Moreover, we found that *trans*-RSV (50 μM) by itself was neurotoxic in the rat primary cortical neuron cultures (Fig. 7c), whereas *cis*-RSV protected against the neurotoxicity induced by *trans*-RSV in a dose-dependent manner (Fig. 7d). Furthermore, a high concentration of *trans*-RSV and D-tyrosine increased the levels of cleaved caspase-3 (a marker of apoptosis) in rat primary cortical neurons (Supplementary Fig. 9d, e). In contrast,

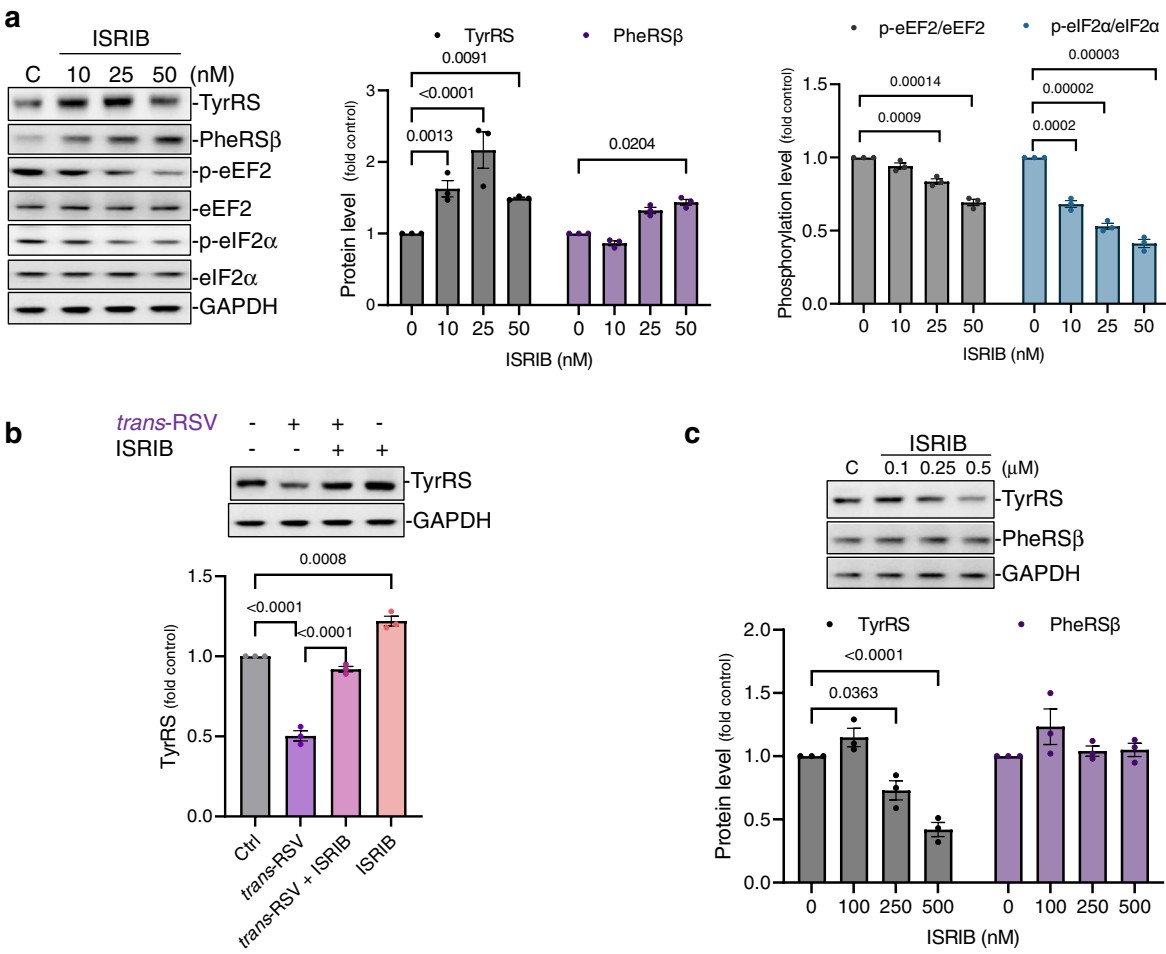

**Fig. 5 ISRIB has concentration-dependent dichotomic effects on neuronal TyrRS levels. a** *Lower doses of ISRIB increase TyrRS protein levels in neurons.*
Primary cortical neurons were treated with ISRIB (≤50 nM) for 8 hr and changes in the levels of TyrRS, PheRSβ, p-eEF2, eEF2, p-eIF2α, and eIF2α were
detected and quantified by WB analysis using their respective antibodies. **b** *Low dose ISRIB protects neurons against trans-RSV-mediated depletion of TyrRS.*
Primary cortical neurons were treated with *trans*-RSV (25 μM) alone or in combination with ISRIB (10 nM) for 8 hr, and changes in the levels of TyrRS were
detected by WB. **c** *High concentrations of ISRIB decrease TyrRS protein levels in cortical neurons.* Primary rat cortical neurons were treated with ISRIB
(100–500 nM) for 8 hr and changes in the levels of TyrRS, and PheRSβ were detected by WB analysis using their respective antibodies. All data represent
mean ± SEM for n = 3 independent experiments with significance measured using two-way ANOVA with Dunnett's test for multiple comparisons.

*cis*-RSV decreased the levels of caspase-3 cleavage (Supplementary Fig. 9d). However, treatment with either ISRIB, eEF2K inhibitor, or DA protected against neurotoxic effects of *trans*-RSV (Supplementary Fig. 9f–h), indicating a critical role of sustained protein synthesis in neuronal DNA repair and survival.

**Cis-RSV and trans-RSV have opposite effects on the auto-PARylation of PARP1 in rat cortical neurons.** Similar to the circadian regulation of tyrosine (Supplementary Fig. 7a), the auto-PARylation of PARP1 is also circadian-regulated[29]. We previously showed that tyrosine inhibits the auto-PARylation of PARP1[28] while *cis*-RSV induces a 'tyrosine-free' conformation in TyrRS[25] to stimulate the auto-PARylation[28]. Because *trans*-RSV mimics the 'tyrosine-like' conformation in TyrRS[28], we analyzed the effects of *cis*- and *trans*-RSV on the auto-PARylation of PARP1[28]. As expected, a low concentration of *trans*-RSV (5 μM), which converts to *cis*-RSV in solution[27,28], and *cis*-RSV stimulated the auto-PARylation of PARP1 (Fig. 7e, f), whereas higher concentrations of *trans*-RSV (≥25 μM) inhibited the auto-PARylation of PARP1 (Fig. 7f). Interestingly, the apparent Ki value of *trans*-RSV-mediated inhibition of tyrosine activation by TyrRS in an ATP-PPi exchange assay was ~25 μM[28] which is the lowest concentration of *trans*-RSV that significantly inhibits the

auto-PARylation of PARP1 (Fig. 7f). In addition, we previously showed that tyrosine inhibits auto-PARylation-dependent acetylation of proteins[28]. Further, 500 mg/day dosing of *trans*-RSV (68.5 μM plasma level) inhibited the acetylation of H3 at lysine 56 (AcK56-H3) in humans[23]. Consistently, *trans*-RSV (≥25 μM) inhibited H3 and H4 acetylation at lysines 56 and 16 respectively (AcK56-H3 and AcK16-H4) (Fig. 7e, f and Supplementary Fig. 9i, j), whereas *cis*-RSV increased the acetylation of H4 lysine 16 (AcK16-H4). Furthermore, TyrRS knockdown diminished the effects of low-concentration (5 μM) *trans*-RSV and *cis*-RSV (25 μM) on the auto-PARylation of PARP1 (Fig. 7g and Supplementary Fig. 9k), supporting our previous findings using low-concentration (≤5 μM) *trans*-RSV[28].

**Cis-RSV stimulates the de-ADP-ribosylation of chromatin in the rat cortical neurons.** Although HPF1 and serine-ADP-ribosylation are decreased in AD brains (Fig. 6a), PARP1 can also PARylate on the glutamic/aspartic acid (Glu/Asp) residues of its substrates in the presence of broken DNA. Consistent with the accumulation of neuronal DNA damage[51], the brain samples of AD patients show increased levels of nuclear PARylation[53], suggesting a potential role of DNA damage-induced PARylation in neurons. Auto-PARylation dissociates PARP1 from the

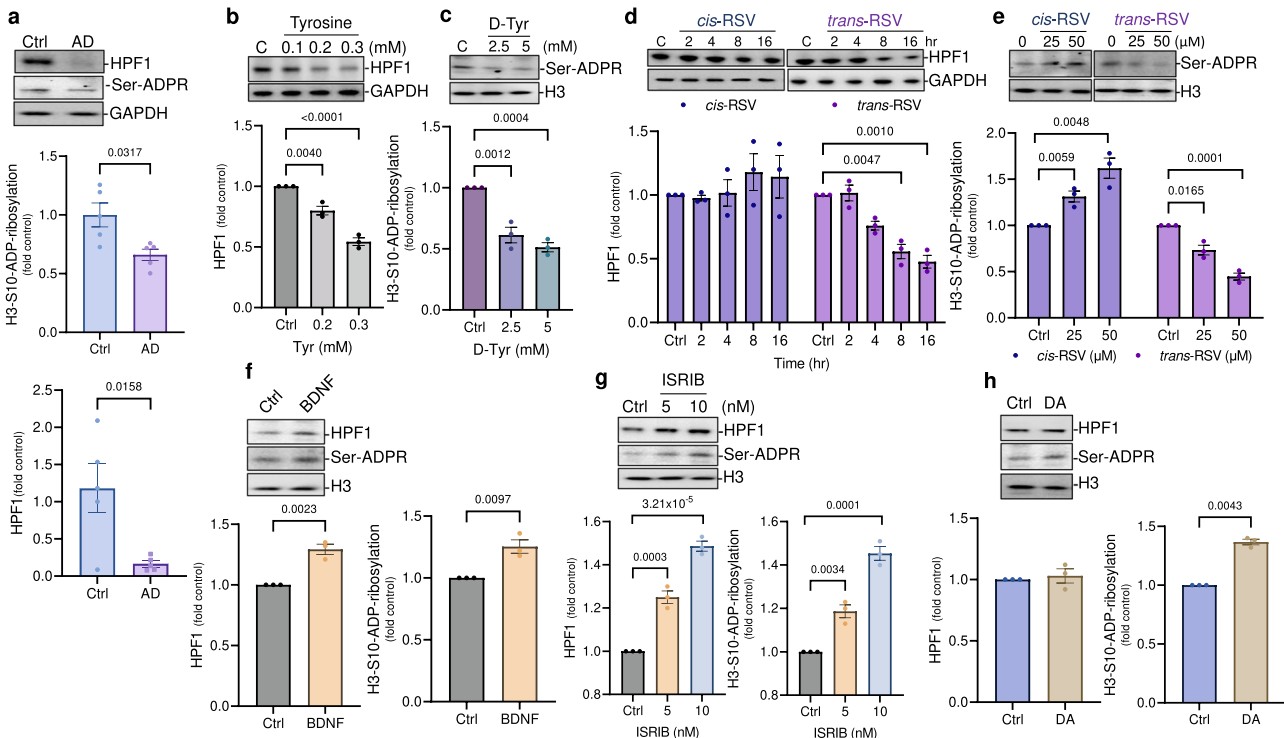

**Fig. 6 Histone serine-ADP-ribosylation is decreased in AD brains and *cis*- and *trans*-RSV have opposite effects on it. a** *HPF1 and histone serine-ADP-ribosylation are decreased in the hippocampal region of AD patients.* Representative immunoblots and quantification for HPF1 and histone H3-Ser10-ADP-ribosylation (Serine-ADPR) using anti-HPF1 and anti-H3-Ser10-ADPR antibodies, respectively, in the hippocampal region of AD patients ($n = 5$) with age and sex-matched controls ($n = 5$). **b** *Tyrosine decreases the protein levels of HPF1.* Representative immunoblots and quantifications for HPF1 in rat cortical neurons (DIV 9/10) after treatment with Tyr ($\leq$300 μM) for up to 8 h for $n = 3$ experiments. **c** *D-tyrosine inhibits serine-ADP-ribosylation.* Representative immunoblots and quantification for histone H3-Ser10-ADP-ribosylation levels in cortical neurons (DIV 9) using anti-H3-Ser10-ADPR antibody after treatment with D-tyrosine (2.5 and 5 mM). **d**, **e** *cis-and trans-RSV have opposite effects on HPF1 and histone serine-ADP-ribosylation levels.* Representative immunoblots and quantification for HPF1 and histone H3-Serine-ADPR using anti-HPF1 and anti-H3-Ser10-ADPR antibodies, respectively in cortical neurons after treatment with *cis-*and *trans*-RSV (25–50 μM). **f**, **g** *BDNF and ISRIB increase HPF1 levels along with the activation of histone serine-ADP-ribosylation.* Representative immunoblots and quantification for HPF1 and histone H3-Serine-ADPR using anti-HPF1 and anti-H3-Serine-ADPR antibodies, respectively in cortical neurons after treatment with BDNF (50 nM) and ISRIB (5 and 10 nM). **h** *DA activates histone serine-ADP-ribosylation.* Representative immunoblots and quantification for histone H3-Ser10-ADPR levels in cortical neurons (DIV 9) using anti-H3-Ser10-ADPR antibody after treatment with DA (250 μM) for 5–10 min. All data represent mean ± SEM for $n = 3$ experiments with significance measured using ANOVA with Dunnett's test for multiple comparisons.

chromatin and inhibits its activity, while the sustained presence of PARP1 on the chromatin may trigger neurotoxicity[30]. Because *trans*-RSV inhibited the auto-PARylation (Fig. 7f) and depleted HPF1 along with serine-ADP-ribosylation (Fig. 6d, e), we hypothesized that *trans*-RSV would increase PARP1-dependent *trans*-PARylation of the chromatin. As expected, *trans*-RSV increased the association of PARP1 with the chromatin and increased the levels of PARylated proteins in the chromatin fraction (Fig. 8a), which can be potentially considered as reminiscent of increased nuclear PARylation in AD brains[53]. Further, low concentrations of *trans*-RSV ($\leq$10 μM) and *cis*-RSV prevented the interaction of PARP1 with histone H3 while the higher concentrations ($\geq$25 μM) of *trans*-RSV increased it (Supplementary Fig. 10a). Although *cis*-RSV-mediated auto-PARylation of PARP1 resulted in its removal from the chromatin, unexpectedly, we found that *cis*-RSV activated the de-ADP-ribosylation of the chromatin fraction along with higher levels of TyrRS (Fig. 8a). ADP-ribosyl-acceptor hydrolase 3 (ARH3) removes nuclear PARylation, and as expected, *cis*- and *trans*-RSV had differential effects on the recruitment of ARH3 to the chromatin (Fig. 8a). Despite having increased levels of nuclear PARylation[53], the levels of ARH3 remained unchanged in the hippocampal region of human AD patients (Fig. 8b), indicating that ARH3 may not be functional in human

AD brain tissues in the absence of TyrRS. Consistently, we found that TyrRS interacted with ARH3 (Fig. 8c), suggesting a role of TyrRS in the removal of nuclear PARylation (Fig. 8a) that enhances neuronal DNA repair and survival.

**'Trapped' PARP1 inhibits DNA repair and mediates the neurotoxic effects of *trans*-RSV in rat cortical neurons**. Suicidal crosslinking of PARP1 to the damaged DNA causes neurotoxicity[54], and therefore, cell survival depends on removing 'trapped' PARP1 from the broken DNA by either ablation or auto-PARylation[30]. Because *trans*-RSV caused DNA damage (Fig. 5a, b) and inhibited the auto-PARylation of PARP1 (Fig. 7f), and ablation of PARP1 rescues 'trapped' PARP1-mediated neurotoxicity[54], we hypothesized that *trans*-RSV-mediated neurotoxicity is exerted through 'trapped' PARP1 on the damaged DNA. Consistently, small interfering RNA (siRNA) knockdown of PARP1 (siRNA^PARP1) protected against *trans*-RSV-mediated neurotoxicity but did not interfere with the effect of *cis*-RSV (Fig. 8d). These results indicate that while 'trapped' PARP1 mediates the neurotoxic effects of *trans*-RSV, *cis*-RSV effects are not dependent on PARP1 alone. Consistently, we found that *cis*-RSV facilitated the recruitment of PARP2 along with other

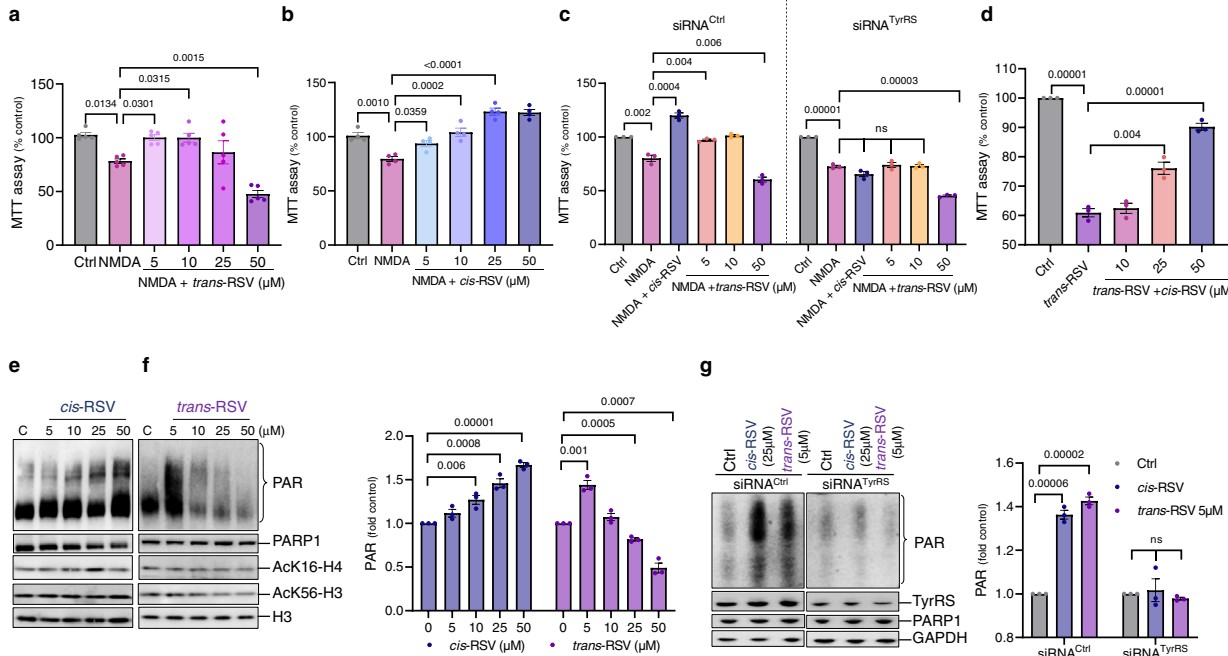

**Fig. 7 Cis- and trans-RSV have opposite effects on the auto-PARylation of PARP1 and neuronal survival under stress. a** *Trans-RSV exacerbates NMDA-mediated neurotoxicity*. Rat cortical neurons (DIV 9) were treated with NMDA (50 µM for 5 min) and then with *trans*-RSV (5–50 µM) for 24 hr. Cells were then exposed to NMDA (500 µM for 5 min), and viability was assessed using MTT assay after 24 hr. Data represents $n = 5$ experiments. **b** *Cis-RSV provides dose-dependent neuroprotection against NMDA-mediated neurotoxicity*. Rat cortical neurons (DIV 9) were treated with NMDA (50 µM for 5 min) and then with *cis*-RSV (5–50 µM) for 24 hr. Cells were then exposed to NMDA (500 µM for 5 min), and viability was assessed using MTT assay after 24 hr. Data represents $n = 4$ experiments. **c** *TyrRS knockdown blunts the neuroprotective effects of cis-RSV and exacerbates the neurotoxicity of trans-RSV*. Rat cortical neurons (DIV 7) were transfected with TyrRS or control siRNA (75 nM) and then treated with *cis*-RSV (50 µM) or *trans*-RSV (5, 10, 50 µM) for 24 hr. Neurons were then exposed to excitotoxic NMDA (500 µM for 5 min), and viability was assessed using MTT assay after 24 hr. **d** *cis-RSV protects from trans-RSV-mediated neurotoxicity*. Rat cortical neurons (DIV 8) were treated with *trans*-RSV alone or combined with different doses of *cis*-RSV (10–50 µM) for 48 hr, and viability was measured using MTT assay. **e**, **f** *Cis-RSV and trans-RSV have opposite effects on the auto-PARylation of PARP1*. Representative immunoblot images and quantification using specific antibodies for PARylation, PARP1, AcK16-H4, AcK56-H3 levels after treatment of cortical neurons (DIV 9) with *cis*- and *trans*-RSV for 15 min. **g** *cis-RSV and low dose trans-RSV-dependent auto-PARylation of PARP1 is TyrRS dependent*. Rat cortical neurons (DIV 7) were transfected with control and TyrRS siRNA followed by treatment with *cis*- (25 µM) and *trans*-RSV (5 µM) for 15 min and immunoblotting and quantification using the specific antibodies for PARylation, PARP1 and TyrRS. All data represent mean ± SEM for $n = 3$ experiments with significance measured using ANOVA with Dunnett's test for multiple comparisons.

DNA repair factors such as HPF1 and OGG1 to the chromatin while *trans*-RSV prevented their recruitment (Supplementary Fig. 10b). However, PARP1 inhibits flap endonuclease (FEN1)-dependent long patch base excision repair (LP-BER), and consistently, *cis*-RSV decreased the recruitment of FEN1 (Supplementary Fig. 10b), indicating a potential role of TyrRS/ARH3-dependent de-ADP-ribosylation of PARP1 in limiting long patch BER. As further evidence of DNA repair, we assessed whether DNA synthesis occurred following *cis*- or *trans*-RSV treatment. The incorporation of nucleoside analogs into DNA from non-dividing neurons was previously used as a readout of neuronal DNA repair[55] and depletion of PARP1 using siRNA or inhibition of PARP1 using small molecules increases the incorporation of nucleoside analogs[55]. In agreement with the observation that 'trapped' PARP1 on the DNA impairs BER[30], *trans*-RSV prevented the incorporation of the nucleoside analog CldU into DNA fibers isolated from neurons, suggesting that DNA-repair associated synthesis is severely inhibited by *trans*-RSV (Fig. 8e and Supplementary Fig. 10c). In contrast, CldU incorporation after *cis*-RSV treatment was detectable albeit reduced by 15–20% compared to control (Fig. 8e and Supplementary Fig. 10c), suggesting the activation of PARP1-dependent short patch BER (SP-BER)[55], potentially through the displacement of FEN1 (Supplementary Fig. 10b) and/or increased recruitment of unmodified

PARP1 to the chromatin (Fig. 8a) that limits DNA resection/repair[56].

**PARP inhibitors are neurotoxic in HR-deficient post-mitotic rat cortical neurons.** Post-mitotic neurons are homologous recombination (HR)-deficient[57] and utilize non-homologous end-joining (NHEJ) for DNA repair. Interestingly, H3 serine-ADP-ribosylation facilitates NHEJ[58] and Ku-dependent DNA repair is inhibited in AD brains[51]. While PARP1 depletion increases HR[56], PARP inhibitors drive toxic NHEJ in HR-deficient cells in an ataxia-telangiectasia mutated (ATM)-dependent manner[59]. Because we previously showed that TyrRS activates ATM through acetylation[28] and *cis*-and *trans*-RSV have opposite effects on TyrRS levels (Fig. 3a, b) and the auto-PARylation of PARP1 (Fig. 7e, f), we tested the effect of well-known PARP inhibitors on *cis*-and *trans*-RSV-mediated effects on neurons. While treatment with the PARP1-specific inhibitor AG-14361 (AG) did not affect *cis*-RSV-mediated neuroprotective effects, treatment with olaparib (Ola) that inhibits both PARP1 and 2 mitigated the neuroprotective effects of *cis*-RSV (Fig. 9a) and did not affect *trans*-RSV-mediated neurotoxicity (Fig. 9b). Because siRNA[PARP1] mitigated the effect of *trans*-RSV (Fig. 8d), taken together, these results indicate a critical neuroprotective role of PARP2, which may be utilized by *cis*-RSV in the absence of PARP1. Moreover, we found that PARP inhibitors themselves

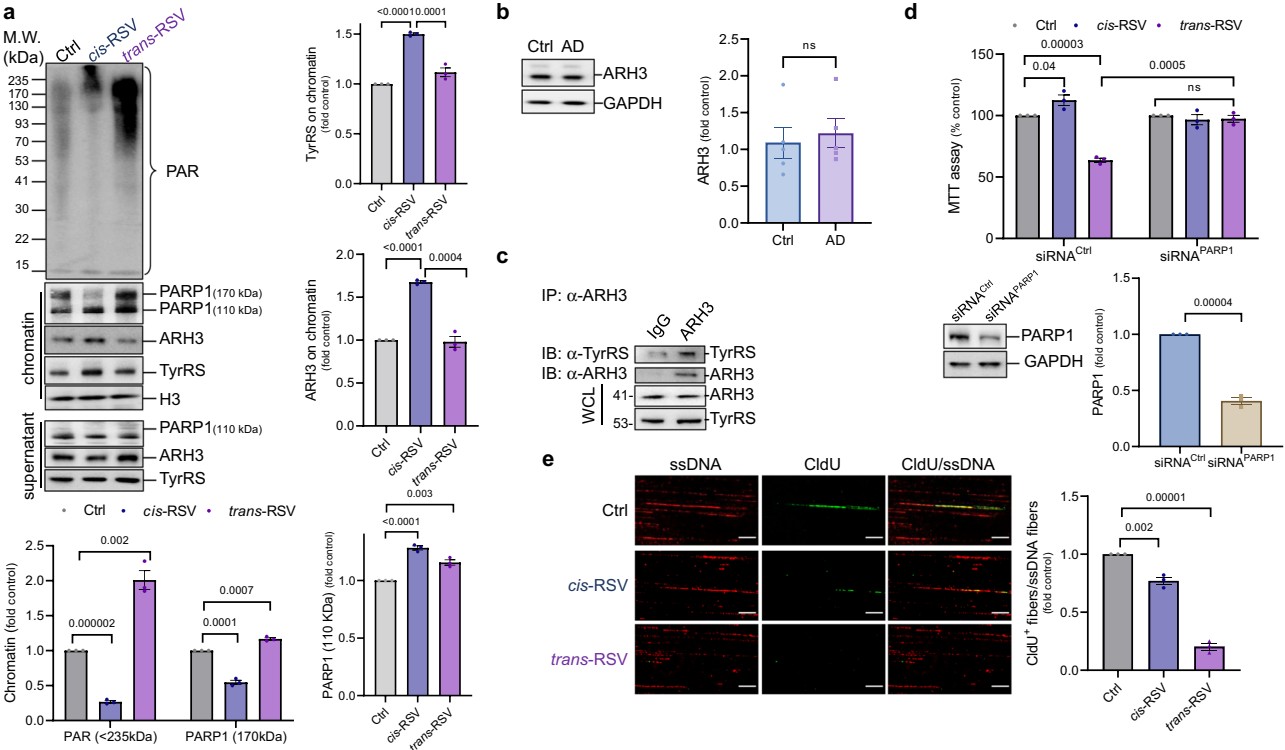

**Fig. 8 Cis- and trans-RSV have opposite effects on ARH3-mediated chromatin de-ADP-ribosylation and 'trapped'-PARP1 mediates the neurotoxic effects of trans-RSV. a** *Cis-RSV removes auto-PARylated PARP1 from chromatin and trans-RSV 'traps' PARP1 onto the chromatin.* Representative immunoblots and quantification from chromatin fraction of cortical neurons (DIV 9) depicting PARP1 and PAR, ARH3, TyrRS after treatment with cis- and trans-RSV (50 µM) for 1 hr. **b** *ARH3 levels are not affected in the hippocampal tissues of AD patients.* Representative immunoblots and quantification for ARH3 using anti-ARH3 antibody in the hippocampal region of AD patients (n = 5) with age and sex-matched controls (n = 5). **c** *TyrRS interacts with ARH3.* Immunoprecipitated (IP) ARH3 from cortical neurons (DIV 9) was immunoblotted (IB) using anti-TyrRS and anti-ARH3 antibodies to detect the interaction of TyrRS with ARH3. Total TyrRS and ARH3 were detected by WB in the whole cell lysate (WCL). **d** *siRNA knockdown of PARP1 rescues trans-RSV-mediated neurotoxicity.* Rat cortical neurons (DIV 7) were transfected with siRNA against PARP1 (siRNA$^{PARP1}$) or control siRNA (75 nM) and then treated with cis-RSV (50 µM) or trans-RSV (50 µM) for 72 hr. Neuronal viability was assessed and quantified using an MTT assay. The knockdown was verified using immunoblot and quantified using specific antibodies for PARP1. **e** *trans-RSV inhibits nucleoside incorporation in a DNA fiber assay.* Cortical neurons (DIV 9/10) were treated with cis-or trans-RSV (50 µM) for 8 hr followed by a 30 min pulse labeling using 50 µM of nucleoside analog, CldU (5-chloro-2'-deoxyuridine). DNA fiber assay was performed according to the published protocol followed by immunostaining for single-stranded (ss) DNA (red) and CldU (green). Representative images (scale bar, 100 µm) showing the incorporation of CldU into DNA during the repair process was assessed using ImageJ by counting the number of CldU positive DNA fibers for a total of 300 fibers per condition. All data represent mean ± SEM with statistical significance calculated using two-tailed Student's unpaired t-test.

decreased neuronal survival (Fig. 9a, b), induced neuronal DNA damage (Fig. 9c) and neurite degeneration (Fig. 9d), suggesting that PARP inhibitors trigger cell death in post-mitotic HR-deficient neurons through the inhibition of H3 serine-ADP-ribosylation-dependent NHEJ[58].

In summary, the mechanism for opposite effects of cis- and trans-RSV on neuronal survival emerging from our studies is illustrated in Fig. 9e. In this model, different forms of stress facilitate the interaction of TyrRS with PARP1/2 leading to their auto-serine-PARylation and subsequent removal from the damaged DNA allowing recruitment of DNA repair factors such as HPF1, ARH3, and OGG1 to repair the damage efficiently. cis-RSV-bound TyrRS facilitates the removal of auto-serine-PARylated PARP1/2 from chromatin while activating ARH3-mediated removal of ADP-ribose to achieve efficient neuronal DNA repair while limiting increased nucleotide incorporation/DNA repair and toxic NHEJ. In contrast, treatment with trans-RSV decreases TyrRS in the absence of which PARP1/2 gets 'trapped' on the damaged DNA and impedes DNA repair leading to subsequent accumulation of DNA damage that drives neurodegeneration.

## Discussion

This study shows that tyrosine is an endogenous negative regulator of TyrRS levels, providing a potential molecular basis for the decreased protein synthesis in the AD brains[31–34], tyrosine-mediated cognitive impairments[8,9] and inhibition of protein synthesis[38,39], axonal degeneration in tyrosinemia patients[10], increased oxidative DNA damage and mutations in aged and AD neurons[50], and the circadian modulation of synaptic TyrRS (Supplementary Fig. 7a)[49]. Interestingly, a mutant amino acid transporter that accumulates tyrosine in *Neurospora crassa* is sensitive to tyrosine[60], and its detoxification is essential for the survival of hematophagous insects[61]. Because synaptic plasticity is regulated at the elongation step, it is conceivable that tyrosine-mediated regulation of TyrRS might be an evolutionary conserved regulatory mechanism of protein synthesis exploited by neurons to enhance plasticity[11]. Since tyrosine level is decreased during the nadir/trough of circadian rhythm[4], our findings might also provide a molecular basis for sleep-stimulated brain protein synthesis[5] and memory formation[6]. Because CR lowers tyrosine levels[15], which are increased during aging (Supplementary Fig. 2 and Supplementary Table 3), our work also provides a molecular

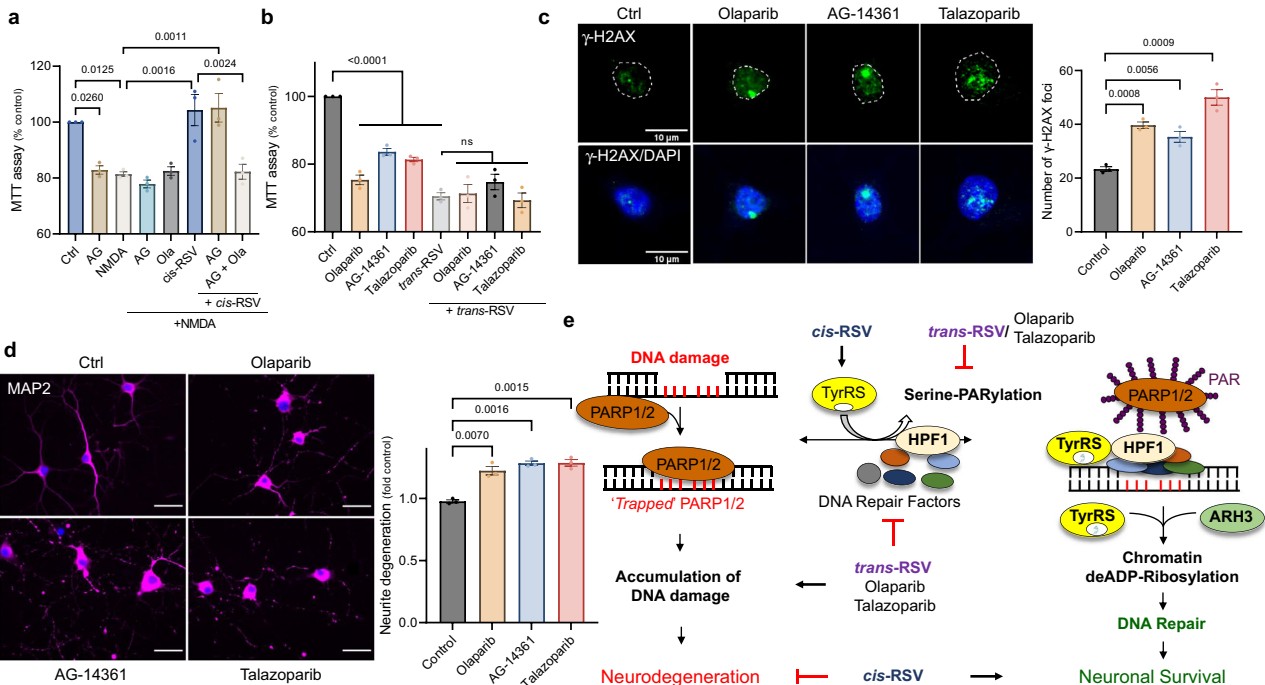

**Fig. 9 PARP inhibitors are neurotoxic in HR-deficient post-mitotic neurons and a potential mechanism of the opposite effects of *cis*-and *trans*-RSV on neuronal survival. a** *PARP is required for cis-RSV-mediated neuroprotective effects*. Rat cortical neurons (DIV 9) were treated with NMDA (50 µM for 5 min), AG-14361 (AG, 10 µM), olaparib (Ola, 10 µM) either alone or in combination with *cis*-RSV (50 µM) for 24 hr. Cells were then exposed to NMDA (500 µM for 5 min) and viability was assessed using MTT assay after 24 hr. **b** *PARP1 inhibitors are neurotoxic and do not affect trans-RSV-mediated neurotoxicity*. Rat cortical neurons (DIV 9) were treated with olaparib (10 µM), AG-14361 (10 µM) and talazoparib (2 µM) either alone or in combination with *trans*-RSV (50 µM) for 72 hr. **c** *PARP1 inhibitors induce DNA damage in HR-deficient post-mitotic neurons*. Immunostaining images (scale bar, 10 µm) for γ-H2AX foci (green; DAPI – nuclear marker, blue) in cortical neurons (DIV 10) after treatment with olaparib (10 µM), AG-14361 (10 µM) and talazoparib (2 µM) for 24 hr. The graph represents the average number of γ-H2AX foci per $n \geq 30$ neurons per treatment condition for $n = 3$ experiments. **d** *PARP1 inhibitors induce neurite degeneration*. Representative images (scale bar, 20 µm) for cortical neurons following olaparib (10 µM), AG-14361 (10 µM) and talazoparib (1 µM) treatment for 24 hr (MAP2 – neurite marker, magenta and DAPI – nuclear marker, blue). Neurons were immunoassayed with anti-MAP2 antibody and quantified for neurite degeneration. **e** *Proposed mechanism of cis-RSV-mediated neuroprotection and trans-RSV-mediated neurotoxicity*. All data represent mean ± SEM with statistical significance calculated using ANOVA with Šídák's test for multiple comparisons.

basis for CR-mediated activation of BER[14] and sleep-mediated activation of neuronal DNA repair[7]. Interestingly, exercise that stimulates the production of BDNF and DA is also known to activate PARP1-dependent DNA repair[62] and protein synthesis in humans[63]. Aging is the single most important contributing factor to the development of AD. Intriguingly, AD does not occur naturally in naked mole rats (NMR), the longest-lived rodents resistant to AD. Because naked mole rats maintain lower serum tyrosine levels[64] and higher levels of auto-PARylation of PARP1, it is tempting to speculate that decreased tyrosine levels in naked mole rats contribute to their longevity and resistance to AD through enhanced TyrRS/PARP1-dependent DNA repair, speculation that will need to be explored in the future. Although increased levels of branched-chain amino acids (BCAAs) are associated with metabolic disorders, in this context, it is interesting to note that BCAAs levels are decreased in AD and ASD. Whether increased levels of tyrosine in children with ASD or mutations of amino acid transporter (LAT1) that increase the tyrosine level in the brain contribute to the increased incidence of mutations and dysregulated protein synthesis in ASD will be of future interest. Since centenarians retain high PARylation levels, which is also required for long-term memory formation[65], these observations suggest that decreased tyrosine levels may be an endogenous stimulator of TyrRS/PARP1-mediated signaling events[28,29], which are dysregulated during aging and in neurocognitive and metabolic disorders. Because PARP1 regulates myelination[66] and tyrosinemia exhibits peripheral neuropathy

and demyelination[10], it will be of future interest to test if mutations of TyrRS that result in dominant-intermediate Charcot-Marie-Tooth neuropathy (DI-CMT)[67] would modulate TyrRS-mediated PARP1 activation[28]. Therefore, tyrosine-mediated depletion of TyrRS (Fig. 1), and tyrosine-mediated induction of 8-oxo-dG, γH2AX, and DNA damage (Fig. 4) shown here may modulate human aging and exacerbate motor, cognitive, and metabolic disorders (Fig. 10).

Our study also provides a potential molecular explanation for high dose *trans*-RSV-mediated brain volume loss in AD patients[21], worsening memory in schizophrenia[22], and increased the CVD risk[24], similar to high concentrations of *trans*-RSV that depletes TyrRS and exacerbates neurotoxic effects in our study. On the other hand, low-dose RSV studies that reported beneficial cognitive benefits in postmenopausal women[18] and protected against human heart failure[19] used 50–75 mg twice a day dose of RSV (i.e; 6–9 µM RSV/dose) similar to low concentrations of *trans*-RSV that, like *cis*-RSV, increased TyrRS and provided neuroprotective effects in our study. Consistently, oxidative DNA damage is elevated in CVD, and PARP1-dependent DNA repair is inhibited in mice models of heart failure[68]. Because inflammation inhibits PARP1-dependent DNA repair[69] and PARP1 modulates chromatin modification and gene expression, potentially regulating myelination[66], CaMKII-dependent neurogenic program[70], and long-term memory formation[65], future studies are required to determine if these functions of PARP1 are affected in AD brains and contribute to *cis*-and *trans*-RSV-mediated

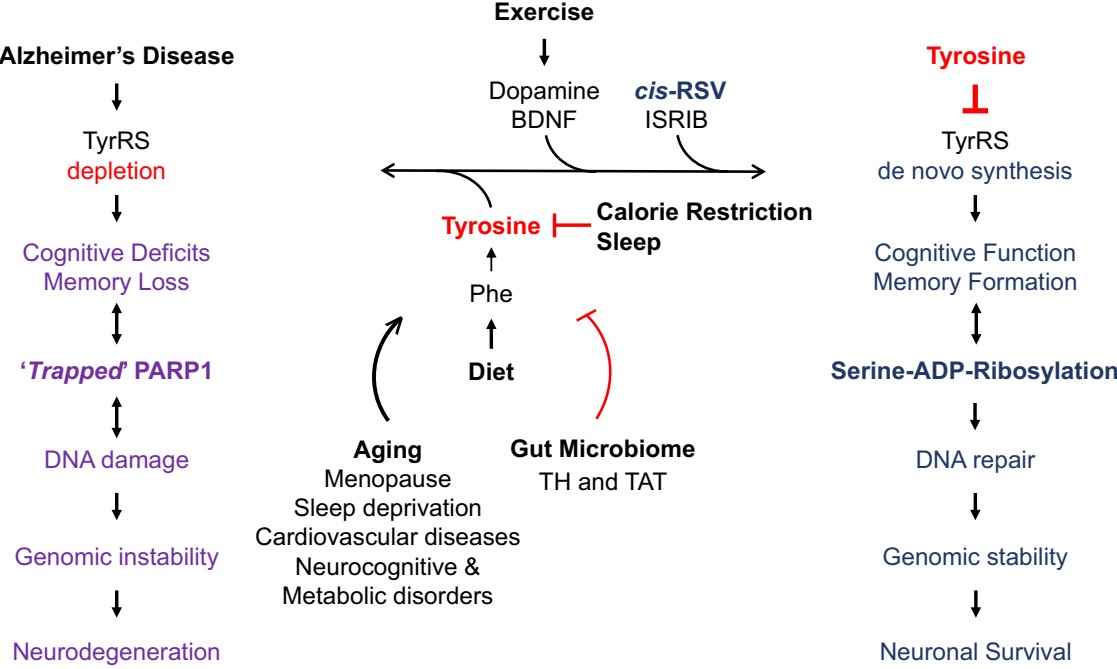

**Fig. 10 Potential mechanism of tyrosine-mediated induction of age-associated neurocognitive disorders, DNA damage, genomic instability, and neurodegeneration.** Age-associated increase in serum tyrosine levels may decrease the de novo synthesis of neuronal TyrRS. Serum tyrosine levels may be modulated by changes in lifestyle like diet, exercise and sleep or by metabolic disorders. Exercise that increases BDNF and dopamine levels may increase the brain protein levels of TyrRS and histone serine-ADP-ribosylation. Similarly, TyrRS levels may be increased using cis-RSV and low concentration ISRIB. In the absence of TyrRS, PARP1 may remain 'trapped' on the broken DNA resulting in neurotoxicity. Therefore, TyrRS and histone serine-ADPribosylation may play critical role in the regulation of DNA damage response to modulate cognitive performance and memory formation.

neuronal effects shown here. Therefore, in addition to a plausible explanation for the apparent benefits of low doses of *trans*-RSV[16–20] and TyrRS being already nominated as a therapeutic target against AD by the National Institute on Aging's Accelerating Medicines Partnership in Alzheimer's Disease (AMP-AD) consortium (https://agora.adknowledgeportal.org/genes/ (genes-router:gene-details/ENSG00000134684), our study suggests that *cis*-RSV or compounds that use *cis*-RSV conformation as a pharmacophore may help in the chronotherapy of age-associated neurocognitive disorders and potentially degenerative and meta-bolic diseases of other tissues.

## Methods
Our research complies with all relevant ethical regulations by University of South Carolina, University of Washington School of Medicine and Wake Forest School of Medicine.

**Postmortem hippocampal tissue samples**. All human postmortem tissues were obtained from the University of Washington School of Medicine BioRepository and Integrated Neuropathology (BRaIN) Laboratory and Precision Neuropathology Core. AD diagnosis was based on cognitive assessments of dementia and neuropathological verification of AD neuropathologic change using Braak staging (AD stages V–VI) and Consortium to Establish a Registry for Alzheimer's Disease (CERAD) scores[71,72]. Studies were performed using flash-frozen hippocampal tissue from male and female patients clinically diagnosed with AD ($n = 7$) and age-matched controls ($n = 7$). Donor information is presented in Supplementary Table 1. The mean age of death was 89.6 years. Postmortem interval (PMI) ranged between 2 and 11 h, with a mean of 5.3 h. Written informed consents were obtained from the patients before sampling by the Anatomical Gift ACT, Washington Statute Chapter 68.64 RCW. BRaIN Laboratory and Precision Neuropathology Core work with the UW School of Medicine Compliance office on the consent forms and the Health Insurance Portability and Accountability Act (HIPAA) compliance.

**Animals**. All mice were housed at the Wake Forest School of Medicine barrier facility under the supervision of the Animal Research Program. Mice adhered to a 12-hr light/12-hr dark cycle, with regular feeding, cage cleaning, and 24-hr food

and water access. WT and eEF2K$^{+/-}$ mice tissue samples were obtained from a previously published study[73].

**Cell culture**. Primary cortical neurons were dissected from E18 Sprague Dawley rats pups in Hibernate E (BrainBits) and dissociated using the Neural Tissue Dissociation kit (Miltenyi Biotec). Minced cortices were incubated in a pre-warmed enzyme mix at 37 °C for 15 min; tissues were then triturated and strained using a 40 μm cell strainer. After washing and centrifugation, neurons were seeded in 50 μg/ml poly-D-Lysine (Sigma Aldrich) coated tissue culture plates. NBActive-1 medium (BrainBits) supplemented with 100 U/ml of Penicillin-Streptomycin (Life Technologies), 2 mM L-Glutamine (Life Technologies), and 1X N21 supplement (R&D Systems) was used as culture medium. For preparation of culture medium containing reduced tyrosine, the components of NBActive-1 medium were altered to combined with a reduced concentration of tyrosine, as mentioned in Supplementary Table 4, and was supplemented with 100 U/ml of Penicillin-Streptomycin (Life Technologies), 2 mM L-Glutamine (Life Technologies), and 1X N21 supplement (R&D Systems) to obtained culture medium with reduced tyrosine. Control (non-targeting), TyrRS, and PARP1 siRNAs were obtained from Invitrogen (# AM4635, s443, and s130207, respectively). Rat cortical neurons at 5 DIV were transfected with 75 nM control or TyrRS siRNA using Dharmafect 3 Transfection Reagent. A second transfection was done two days later using 75 nM of TyrRS siRNA, followed by cell collection or assays after another 48 hr. For PARP1 siRNA, neurons at 7 DIV were transfected with 75 nM siRNA for both control and PARP1 siRNA.

**Puromycin incorporation assay**. Puromycin (10 μM) was added in the last 15 min of the pharmacological treatment on rat cortical neurons (DIV 8). The samples were lysed, followed by western blotting. Puromycin-labeled proteins were identified using the mouse monoclonal antibody 12D10 (1:5000; EMD Millipore, catalog MABE343). Protein synthesis levels were determined by analyzing total lane density from 10 kDa to 250 kDa. Densitometric analysis was performed using ImageJ (Version 1.53c).

**Immunoprecipitation (IP) and Western blot analysis**. For immunoprecipitation, cortical neurons were lysed with a mild lysis buffer (20 mM Tris-HCl (pH 7.5), 150 mM NaCl, 1 mM Na$_2$EDTA, 1 mM EGTA, 1% Triton, 2.5 mM sodium pyrophosphate, 1 mM beta-glycerophosphate, 1 mM Na$_3$VO$_4$, 1 μg/ml leupeptin) containing protease-inhibitor cocktail (Roche, Cat# 04693124001) on ice for 10 min, sonicated with 1 s on/2 s off for 10 s by 30% amplitude of Sonic dismembrator Model 50 (Fisher Scientific), and centrifuged with 12,000 g for 10 min.

The supernatant was incubated with 2 µg of primary antibody (eEF2, ARH3, or non-immune immunoglobulin-G) for 1 hr with rotation at 4 °C, followed by incubation with 25 µl Protein G (Recombinant Protein G Agarose, Invitrogen) beads (pretreated with 10 mg/ml BSA) for 30 min with rotation at 4 °C. Immunoprecipitates were washed three times with lysis buffer and boiled in 1xSDS buffer. For western blot analysis, cultured neurons (DIV 9/10) were washed twice with cold 1× PBS and lysed in cell lysis buffer (20 mM Tris-HCl (pH 7.5), 150 mM NaCl, 1 mM Na₂EDTA, 1 mM EGTA, 1% Triton, 2.5 mM sodium pyrophosphate, 1 mM beta-glycerophosphate, 1 mM Na₃VO₄, 1 µg/ml leupeptin supplemented with protease inhibitor). The lysates were centrifuged at 12,000 g for 15 min at 4 °C to separate the chromatin-bound and soluble fractions. Lysates were quantified using Bio-Rad Protein Assay, and equal amounts of protein were loaded onto a 4 to 12% gradient gel (NuPAGE-Invitrogen). Protein was transferred from the gel to 0.2 µm NC membranes at 25 V for 10 min using transfer stacks (iBlot 2- Invitrogen) and blocked with 5% non-fat milk in TBST (10 mM Tris-HCl pH 8.0, 150 mM NaCl, 0.01% Tween-20) for 1 hr before application of primary antibodies. Primary and secondary antibodies were incubated overnight at 4 °C and for 1 hr at room temperature, respectively. Immobilon ECL Ultra Western HRP Substrate (WBULS0500, Millipore) and a luminescent image analyzer (ChemiDoc Imaging System, Bio-Rad) were used to detect proteins. Quantification of western blots was done using ImageJ (Version 1.53c).

**List of antibodies used for western blotting**

| Antibody | Company | Catalog No. | Dilution |
|---|---|---|---|
| Acetyl-Histone H3 (Lys56) | Cell Signaling Technology | 4243 | 1:1000 (WB) |
| Acetyl-Histone H4 (Lys16) | Cell Signaling Technology | 13534 | 1:1000 (WB) |
| ARH3 | Proteintech | 16504-1-AP | 1:1000 (WB) |
| ATF4 | Cell Signaling Technology | 11815 | 1:1000 (WB) |
| Cleaved caspase 3 | Cell Signaling Technology | 9664 | 1:1000 (WB) |
| eEF2 | Cell Signaling Technology | 2332 | 1:1000 (WB) |
| eIF2α | Cell Signaling Technology | 5324 | 1:1000 (WB) |
| Fen1 | Proteintech | 14768-1-AP | 1:1000 (WB) |
| GAPDH | Cell Signaling Technology | 2118 | 1:2000 (WB) |
| H3 | Proteintech | 17168-1-AP | 1:1000 (WB) |
| H3-S10-ADP-Ribose | Bio-RAD | HCA357 | 1:1000 (WB) |
| H4 | Proteintech | 16047-1-AP | 1:1000 (WB) |
| HPF1 | Novus Biologicals | NBP1-93973 | 1:1000 (WB) |
| OGG1 | Proteintech | 15125-1-AP | 1:1000 (WB) |
| PARP1 | Proteintech | 66520-1-Ig | 1:1000 (WB) |
| PARP2 | Abcam | ab177529 | 1:500 (WB) |
| PheRSα | Proteintech | 18121-1-AP | 1:1000 (WB) |
| PheRSβ | Proteintech | 16341-1-AP | 1:1000 (WB) |
| Phospho-eEF2 (Thr56) | Cell Signaling Technology | 2331 | 1:1000 (WB) |
| Phospho-eIF2α (Ser51) | Cell Signaling Technology | 3398 | 1:1000 (WB) |
| Poly (ADP-Ribose) Polymer | Abcam | ab14459 | 1:1000 (WB) |
| PP2A C | Cell Signaling Technology | 2038 | 1:1000 (WB) |
| Puromycin | Millipore | MABE343 | 1:1000 (WB) |
| TyrRS | Abcam | ab50961 | 1:1000 (WB) |
| β-Tubulin | Cell Signaling Technology | 2128 | 1:2000 (WB) |

**Comet assay**. Cultured cortical neurons (DIV 9) were treated with cis- or trans-RSV (50 µM), in combination with tyrosine (1 mM) for 1 hr. The cells were harvested with a cell scraper using chilled PBS and counted. The comet assay (Trevigen Inc, Gaithersburg, MD) was performed according to manufacturer's protocol using alkaline conditions. Electrophoresis was carried out at the rate of 1.0 V/cm for 20 min. The slides were removed from the electrophoresis chamber, washed twice in deionized water for 5 min and immersed in 70% ethanol for 5 min. Subsequently, the slides were dried at 37 °C for 30 min, DNA was stained with 50 µl of SYBR Gold dye (Fisher Scientific, 1:10000 in Tris–EDTA buffer, pH 7.5) for 20 min in the dark at room temperature and then analyzed using an epifluorescent microscope at 10X magnification. The images were scored for comet parameters, such as tail length and tail moment (product of % of DNA in the tail and tail length) using the Tritek CometScore™ Freeware v1.5 image analysis software.

**DNA fiber analysis**. Cultured cortical neurons (DIV 9) were treated with cis- and trans-RSV (50 µM) for 8 hr, followed by the addition of thymidine analog, 50 µM CldU (5-Chloro-2′-deoxyuridine) for 30 min. DNA fiber analysis was done according to established protocols[74,75]. Briefly, cells were isolated by trypsinization, embedded in agarose plugs, and subjected to proteinase K (0.5% SDS, 0.1 M EDTA, 1 mg/ml Proteinase K) digestion at 50 °C for 16 hr. Plugs were dissolved with agarose (Fisher [NEB], 50-811-726) for 16 hr. Molecular combing was performed using the FiberComb® Molecular Combing System (Genomic Vision) with a constant stretching factor of 2 kb/µm using vinylsilane coverslips (20 × 20 mm; Genomic Vision), according to the manufacturer's instructions. Combed coverslips were incubated at 60 °C for 2 hr in a pre-warmed hybridization oven to minimize photo-breaking, followed by denaturation of the DNA fibers (0.5 M NaOH + 1 M NaCl) for 8 min. The coverslips were then washed with PBS, followed by serial ethanol dehydration (70–100%). Following two 1x PBS washes, the coverslips were blocked in 3% BSA/1x PBS for 30 min followed by incubation with α-BrdU (for CldU) (BD Biosciences 347580) (1:100) and ssDNA antibody (Millipore MAB3034) (1:100), for 2 hr at 37 °C. After three PBST washes, secondary antibody incubation was done using α-mouse AlexaFluor 594 and α-rat AlexaFluor 488 (1:100) for 1 hr at 37 °C. Coverslips were washed three times with 1x PBST, dehydrated and mounted on slides with mounting media. The stained DNA fibers were visualized using a fluorescence microscope (EVOS FL, ThermoFisher Scientific). Analysis was performed in ImageJ by counting the total ssDNA (red) and the CldU labeled fibers (green). For each treatment condition, 300 fibers were counted, and the average ratio of CldU incorporation for ssDNA fibers per condition was used for final representation.

**Immunofluorescence (IF)**. Cultured cortical neurons (DIV 9/10) were fixed in 4% formaldehyde for 15 min, followed by permeabilization and blocking for 30 min in 5% BSA (PBS) and 0.1% (Tween20) at room temperature. Incubation with primary antibodies was done at 4 °C overnight. The names and dilutions of the primary antibodies used for IF are described below. Secondary antibodies were incubated for 1 hr at room temperature. Secondary antibodies used were: Alexa Fluor 647 (anti-chicken), Alexa Fluor 555 (anti-mouse), Alexa Fluor 488 (anti-rabbit) from Invitrogen at 1:1000 dilution. Coverslips were then mounted using DAPI (4′,6-diamidino-2-phenylindole)-supplemented mounting medium, Prolong Gold Antifade (Invitrogen) and imaged with Leica DMI6000 epifluorescent microscope using oil immersion 63x/NA 1.4 objective. The quantification for total protein levels in neurons was done using ImageJ (Version 1.53c), and imaging parameters were matched for exposure, gain, and offset. Neuronal γ-H2AX foci was calculated as shown previously[76].

**List of antibodies used for IF**

| Antibody | Company | Catalog No. | Dilution |
|---|---|---|---|
| MAP2 | Abcam | Ab5392 | 1:500 |
| phospho-histone H2AX (Ser139) | Cell Signaling Technology | 9178 | 1:400 |
| TyrRS | Novus Biologicals | NBP1-32551 | 1:200 |
| 8-hydroxy-2′-deoxyguanosine | Abcam | ab48508 | 1:200 |

**Drug treatments**. All drugs/inhibitors stock solutions (1000x) were prepared in DMSO or ethanol and diluted in culture media for final concentration. cis-RSV was purchased from Cayman Chemicals (Item No. 10004235, ≥ 98% purity) and trans-RSV was purchased from Millipore-Sigma (catalog No. 34092, ≥ 99% purity) and the stocks (100 mM) were prepared in ethanol. The various compounds used for treatments and their stock concentrations are listed below.

| Compound | Catalog | Stock concentration | Final concentration | Solvent |
|---|---|---|---|---|
| 6-OHDA (hydrobromide) | 25330, Cayman Chemical | 100 mM | 0.1–0.3 mM | PBS |
| A484594 | 324516, Millipore | 100 µM | 100 nM | DMSO |
| AG-14361 | A4158, Apex Bio | 10 mM | 10 µM | DMSO |
| Bafilomycin B1 | BVT-004-M001, BioViotica | 100 µM | 100 nM | DMSO |
| BDNF | B3795-5UG, Sigma Aldrich | 100 µg/ml | 100 ng/ml | PBS |
| Cycloheximide | 01810-1 G, Sigma Aldrich | 100 mg/ml | 100 µg/ml | DMSO |
| Dopamine HCl | H60255, Sigma Aldrich | 100 mM | 0.1–0.5 mM | PBS |
| D-Phe | 225200, BTC | 100 mM | 0.5–2 mM | PBS |
| D-Trp | 215145, BTC | 100 mM | 0.1–1 mM | PBS |
| D-Tyr | 143865, BTC | 100 mM | 0.5–2 mM | PBS |

| Etoposide | 28435, Chem Implex | 10 mM | 10 µM | DMSO |
|---|---|---|---|---|
| ISRIB | 5284, Tocris | 250 µM | 5–500 nM | DMSO |
| L-DOPA | A11311, Alfa Aesar | 100 mM | 0.1–0.5 mM | PBS |
| L-Phe | A13238, Alfa Aesar | 100 mM | 0.1–0.5 mM | PBS |
| L-Tyr | 194759, MP Biomedicals | 100 mM | 0.1–0.5 mM | PBS |
| MG132 | 10012628,Cayman Chemical | 100 µM | 100 nM | DMSO |
| MPP$^+$ Iodide | D048, Sigma Aldrich | 100 mM | 50 µM | DMSO |
| Nelfinavir mesylate hydrate | N0986, TCI | 100 µM | 20–40 µM | PBS |
| NMDA | 0114, Tocris | 50 mM | 50 µM | PBS |
| Olaparib | AZD2281, Selleckchem | 10 mM | 10 µM | DMSO |
| Rapamycin | 53123, Alfa Aesar | 100 µM | 5–50 nM | DMSO |
| Talazoparib | A4153, Apex Bio | 10 mM | 1–10 µM | DMSO |

**Neurite degeneration index**. The neurite degeneration index was calculated as described previously[77,78]. Samples were imaged using ImageXpress Micro 4 at a magnification of 10x to capture the entire field of interest. The samples analyzed for neurite degeneration were stained using the standard immunofluorescence procedure with MAP2 (Alexa fluor 647) for neurites and DAPI staining for the nucleus. Neurite degeneration was quantified using 5–6 regions of interest of equal sizes from each treatment condition. The analysis of neurite degeneration was done using ImageJ. The fluorescent images for MAP2 staining were binarized. The pixel intensity of regions corresponding to neurite staining was converted to black, and all other regions were converted to white. Healthy intact neurites show a continuous tract, whereas degenerated axons have a particulate structure due to fragmentation or beading. To detect degenerated neurites, we used the particle analyzer module of ImageJ. We calculated the percentage of the area of the small fragments or particles (size = 3–10 µm$^2$) to the intact neurites (size >25 µm$^2$) with information derived from the binary images. A degeneration index (DI) was calculated as the fragmented neurite area ratio over the intact neurite area.

**Cell viability assays**. Rat cortical neurons (DIV 9/11) were exposed to different treatments (NMDA, ETO, H$_2$O$_2$, MPP$^+$) after seeding 20,000 cells/well in 96-well plates. Cell viability was then assessed 48 hr after the initial NMDA exposure. 3-[4,5-dimethylthiazole-2-yl]-2,5-diphenyltetrazolium bromide (MTT) assays were used to assess cell viability changes. Rat cortical neurons (DIV 9) were exposed to 5 µM etoposide (ETO, 28435 Chem Implex), 400 µM H$_2$O$_2$ (H1009, Sigma Aldrich) or 10 µM MPP$^+$ (D048, Sigma Aldrich) for 24 hr after pre-treatment with cis-RSV or trans-RSV (50 µM) for 16 hr. Cultured rat cortical neurons were incubated with MTT (0.5 mg/mL). In the MTT assay, after 2 hr incubation, the insoluble purple product formazan resulting from the reduction of MTT by NAD(P)H-dependent oxidoreductases present in cells with viable mitochondria was solubilized in dimethyl sulfoxide at room temperature, under agitation, and protected from light. The percentage of MTT reduced as measured by the difference between the absorbances at 570 nm read in a spectrophotometer (Spectramax 190 R Molecular Devices, UK). Results are presented as a percentage of control (wells incubated with the vehicle).

**Use of publicly available proteomics and metabolomics data for the analysis of TyrRS and tyrosine levels**. The proteomic data for TyrRS in human brain samples were obtained from the public databases as mentioned below. The graphical representation for biweight midcorrelation (BICOR) score of TyrRS protein level in the brain was created by retrieving and analyzing data from a large-scale proteomic database associated with a previously published work in ref. [35]. The published proteomic analysis[35] used label-free mass spectrometry to quantitate the protein levels in the clinical samples of dorsolateral prefrontal cortex (DLPFC) regions of patients with or without AD. The parameters used were: disease status, scored as AD = 2, Asymptomatic AD = 1, Control = 0 ($n = 419$), tau neurofibrillary tangle burden (Braak stage, I-VI according to increasing severity, $n = 419$) and cognitive performance assessed by the Cognitive Abilities Screening Instrument (CASI) score ($n = 56$). Differences in protein levels were assessed by a two-sided Welch's $t$-test and corrected for multiple comparisons to obtain $p$ values. Z-score was measured in terms of standard deviations from the mean. The region-specific information about TyrRS protein levels was retrieved from a recent public brain proteomic database associated with a previously published work in ref. [13]. The log fold change in TyrRS protein levels from six distinct regions from human post-mortem brain of AD cases versus asymptomatic controls, namely, entorhinal cortex (ERC), hippocampus (Hip), cingulate gyrus (CG), sensory cortex (SCx), the motor cortex (MCx) and cerebellum (CER) were identified using mass spectrometry from donors ($n = 9$ AD cases, $n = 9$ asymptomatic controls). Statistical significance was determined using a global false discovery rate (FDR) threshold of 5%, i.e., the largest set of proteins with an average local FDR ≤ 5% were deemed significant.

The circadian protein levels of synaptic TyrRS and circadian levels of serum tyrosine were generated using data from the re-analysis of the mouse circadian proteomic[49] and metabolomic[79] data. Metabolomic data for circadian modulation of serum tyrosine levels in normal and sleep-deprived conditions was obtained after re-analysis of published human circadian metabolomic data[80,81] and HPF1 expression levels were obtained from transcriptomics analysis associated with a previously published study[82].

**Statistics & reproducibility**. All quantitative experiments were carried out in triplicate, and graphs represent average ± standard error mean (SEM). Statistical analysis was performed using unpaired, one-tailed or two-tailed Student's $t$-test or two way ANOVA with multiple comparison analysis done with Tukeys's or Dunnett's test to correct for multiple comparisons. All the p values for statistical analysis are represented in the figures and are also reported in the source data file. No statistical method was used to predetermine sample size. No data were excluded from the analyses. The experiments were not randomized. The Investigators were not blinded to allocation during experiments and outcome assessment.

**Reporting summary**. Further information on research design is available in the Nature Research Reporting Summary linked to this article.

# Data availability
All data needed to evaluate the conclusions in the paper are present in the paper and/or the Supplementary Materials. Source data are provided with this paper. Additional data related to this paper may be requested from the authors. Correspondence and requests for materials should be addressed to Mathew Sajish, Email: mathew2@cop.sc.edu.

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

## Acknowledgements

The authors sincerely appreciate the contributions of Paul Schimmel, Ph.D., Igor Roninson, Ph.D., Jeffery L. Twiss, M.D., Ph.D., and Stephen Cutler, Ph.D. for critically reading the manuscript and making helpful suggestions. The authors thank Amar Kar, Ph.D. and Courtney Buchanan for their help with the rat primary neuronal cultures, Ashita Bhan for helping with some of the western blots, and Drs. Erica Melief and Aimee Schantz for administrative support and facilitating brain donor selection and tissue transfer. Authors acknowledge Microscopy Core of the COBRE Center for Targeted Therapeutics (CTT) and the SC SmartState Center for Childhood Neurotherapeutics at the University of South Carolina for assistance with microscopy and funding from NIH COBRE grant (P20GM109091) and NSF (Award Number: 1755670) and American Cancer Society (ACS)- Institutional Research Grant (IRG –17-179-04) (to M.S.), and R01 AG055581, and R01 AG056622 (to T.M.). The authors further appreciate NIH support for the University of Washington Alzheimer's Disease Research Center (P50AG005136 and P30AG066509) and the Adult Changes in Thought (ACT) study (U01006781), the Nancy and Buster Alvord Endowment (to C.D.K.), and research brain donors and their loved ones without whom this research would be impossible.

## Author contributions

M.J. designed and performed most of the experiments, and analyzed the data, and cowrote the manuscript. X.Z. and T.M. and C.D.K. provided tissue samples. C.N.R. and M.C.HIII. helped with some of the analysis of the data and western blot analysis. J.M and M.W performed the comet assay and J.S. performed the DNA fiber assay and helped M.J. with the analysis. M.S. supervised the research, conceived the idea and designed experiments and analyzed the data and wrote the manuscript. All authors reviewed, edited and approved the final draft of the manuscript.

## Competing interests

The authors declare no competing interests.
