## [Peer Review File · Nature Communications]

Reviewers' Comments:

Reviewer #1:

Remarks to the Author:

Jhanji and colleagues report a mechanism for the perturbed activity of Tyrosyl-tRNA synthetase (TyrRS) in the affected brain regions of AD patients. They report that Tyr levels decrease the nuclear and neurite levels of TyrRS in neurons and cause oxidative DNA damage. The authors suggest that levels may have causal effects in human aging and not just in AD models. I will only comment on the DNA repair components of the work as I am sure other experts will cover the other aspects.

1. The authors used γ -H2AX as a proxy for DNA damage (Figure 3a and 3b), which will also mark regions of single stranded DNA. The authors should provide direct evidence for DNA damage by measuring DNA breaks using techniques such as the comet assay.
2. I am not convinced by the role of PARP1, can the authors provide increased serine-PARylation of target proteins in the affected brain regions?
3. How does the auto-PARylation of PARP1 that is circadian regulated correlate with DNA damage? PARP1 got its fingers in many pies, how can the authors ascertain these effects to DNA damage and not, for example, to chromatin modification and gene expression?
4. There is a confusion between ADP-ribose scars and what the authors describe/observe in their study. The scars that mark sites of DNA damage are mono-ADP-ribose and not poly-ADP-ribose.
5. The reduction of HPF1 in hippocampal tissues from AD patients is intriguing. Is it a transcriptional downregulation in response to senescence caused by accumulation of DNA damage? Does the DNA damage accumulate in the HPF1 promoter? The authors need to provide further mechanistic insight to explain this observation.
6. Why does cis-RSV not affect the recruitment of XRCC1 despite its crucial role in PARP1 mediated response to oxidative damage (Suppl Fig 8e)?

Reviewer #2:

Remarks to the Author:

Jhanji et al., assemble evidence from the literature that an increase in systemic Tyrosine is associated with cognitive decline, aging, and neurological disorders. They link this observation with findings that Tyrosine mediates protein levels of TyrRS. This effect can be mimicked by Resveratrol (RSV), which has been previously shown to bind to TyrRS and stabilize either TyrRS' Tyr-bound or Tyr-free conformation, dependent on whether RSV is cis- or in trans-configuration. cis-RSV, acting through TyrRS, has positive effects on neuron survival, even under cell stress, while trans-RSV is toxic for neurons at higher concentrations. Among the cell survival promoting effects of cis-RSV/TyrRS, is DNA damage repair through control of PARP1 auto-PARylation and rescue of trapped PARP1.

Overall, it is a beautiful manuscript of high relevance due to the ease with which the development of therapeutics and small molecules can be seen to evolve from this study. The presentation is clear, and the data is convincing. I strongly recommend its publication.

The following points could be addressed to aid with the logical flow of the manuscript. Most comments can be resolved without new experiments and are to aid discussion:

While the regulation of TyrRS protein levels by Tyrosine is supported by literature, the mechanism through which this happens is not clear. Canonically, TyrRS and other tRNA synthetases are viewed as stable proteins with high expressions levels and long turnover times. This might very well be different in neurons and the authors show convincing evidence for TyrRS and PheRS being dynamically regulated. The time scale of TyrRS reduction is fast - 4 h of treatment with neurotoxic agents is sufficient to induce a loss of total TyrRS as shown in S4. It would be interesting to know the underlying mechanism, for example through treatment with proteasome and autophagy inhibitors or inhibitors of translation, such as cycloheximide. TyrRS could also be secreted. If neurons are too sensitive for treatment with sublethal concentrations of these inhibitors, ubiquitination of TyrRS might be shown by western blot or available mass spectrometry data.

Given that the reduction of TyrRS protein is both central to the narrative, quite convincing from the author's data, and counter-intuitive, it would be great to have insights into the molecular regulation behind it, and I would like to see it at least discussed.

In line with the above question, is the protein synthesis inhibitory effect of Tyrosine due to the reduction of TyrRS? It is again counter-intuitive why elevated levels of an amino acid should be inhibitory for protein synthesis, for which it is a requisite, therefore, it should be spelled out as clearly as possible in the manuscript.

This is an old question in tRNA synthetase research but given the dramatic drop in TyrRS in Figure 2b I do wonder whether protein synthesis would be affected. Are there any indications that that would be the case, for example by comparing with viability of siTyrRS neurons?

What are the reasons for all cited clinical studies focusing on trans-RSV over cis-RSV or a racemate? The authors suggest that beneficial results from these studies are based on small amounts of trans-RSV converting into cis-RSV but are detrimental at high concentrations, where the intracellular conversion to cis-sulfate metabolites might be saturated. Elaborating on the reasons for trans-RSV being preferentially studied previously will help put the new results presented here into context.

The link to eEF2 and eIF2 could be explained better, preferably with a scheme – I assume dopamine does not chemically dephosphorylate eEF2 itself but instead induces its dephosphorylation through a pathway? Was eEF2K chosen as the target of pharmaceutical inhibition because an inhibitor was readily available or due to its specific function in the eEF2 complex?

TyrRS is among the ATF4 target genes that are specifically upregulated in ISR while general translation is globally reduced (Han, ... Kaufman, 2013). It is surprising that TyrRS levels are affected positively by the suppression of ISR – are there indications that the regulation of gene expression in ISR is different in neurons or is it a concentration dependent effect as the authors do see a repression of TyrRS upon higher doses of ISRIB?

Tyrosine induced DNA damage is especially interesting in neurons as they accumulate over time due to the long life of neuronal cells. Together with the findings on removal of ADP-ribose scars, I'm wondering to which extent RSV is able to reverse DNA damage or if it is mostly protective towards the further accumulation of DNA damage.

Minor issues/typos:

It should be mentioned that the official IUPAC name of TyrRS is YARS1.

It could be stated more clearly in the figure legends whether n-numbers refer only to one panel or to the whole figure (for example 1h, mean \pm SEM, n=3 for all figures or only figure 1h?).

Some material and method sections are justified, others are not.

Figure S1F has a typo: Brain instead of Brian

Use of publicly available proteomics data... has spaces between citation and text, which are missing in the rest of the document.

S3f legend lacks a space between "(rapa, 100nM)"

Cell viability assays: surplus space between "16 hr." and "Cultured rat cortical"

Reviewer #3:

Remarks to the Author:

In this manuscript, Jhanji et al. shows that increased tyrosine levels cause neurotoxicity and DNA damage in neurons by modulating Tyrosyl-tRNA synthetase levels. Next, the authors show that trans-Resveratrol induces a neurotoxic high-tyrosine like state with increased DNA damage whereas cis-Resveratrol induces protective low-tyrosine like state, both mediated through their binding to Tyrosyl-tRNA synthetase. Finally, they demonstrate that the increased DNA damage by trans-Resveratrol is mediated by trapped PARP1 on broken ends of DNA. Overall, this is a highly relevant report highlighting the dose and trans vs cis isomer dependent effect of a widely popular anti-aging supplement and I recommend publication after addressing the comments noted below.

- 1) The authors hypothesize lower Tyr may switch the function of a fraction of TyrRS from protein synthesis to DNA repair. Does TyrRS in Tyr-free state translocate to the nucleus if it acts as a DNA repair factor? Can the authors show evidence for this at low Tyr concentrations?
- 2) The hypothesis for the neuroprotective effects of cis-RSV is that it enhances DNA repair through TyrRS. The authors show that treatment with cis-RSV enhances the cell survival after Etoposide treatment (Fig S7a). Can the authors show that this is indeed mediated by enhanced DNA repair by measuring γ H2AX signal after cis-RSV+Etoposide treatment?
- 3) Similar to comment 1, does cis-RSV increase translocation of TyrRS into nucleus to enhance DNA repair.
- 4) The authors show that PARP1 leads to DNA damage and neurotoxicity through inhibition of DNA repair. However, the model figure in Figure 5e tends to indicate that trans-RSV actively induces DNA damage. Please clarify the model figure or clarify in text.

Minor:

- 1) Indicate γ H2AX is green in figure 3a. It's confusing that cis-RSC is also colored green. Perhaps change the color for γ H2AX signal.
- 2) How was 8-oxo-dG level measured, please indicate in figure legend.

We thank the reviewers for their truly excellent suggestions which have improved the paper and hope we have satisfactorily addressed the points raised by the reviewers.

Reviewer's comments are in blue italics below and our responses are in black.

Reviewer #1 (Remarks to the Author):

Jhanji and colleagues report a mechanism for the perturbed activity of Tyrosyl-tRNA synthetase (TyrRS) in the affected brain regions of AD patients. They report that Tyr levels decrease the nuclear and neurite levels of TyrRS in neurons and cause oxidative DNA damage. The authors suggest that levels may have causal effects in human aging and not just in AD models. I will only comment on the DNA repair components of the work as I am sure other experts will cover the other aspects.

1. The authors used g-H2AX as a proxy for DNA damage (Figure 3a and 3b), which will also mark regions of single stranded DNA. The authors should provide direct evidence for DNA damage by measuring DNA breaks using techniques such as the comet assay.

Response: We thank the reviewer for this excellent suggestion. We conducted the comet assay to provide direct evidence for DNA damage. This data is incorporated in the revised manuscript as follows *“To gain direct evidence for DNA damage, we also conducted the comet assay. We treated cortical neurons with trans-RSV and L-tyrosine either alone or in combination and found that both trans-RSV, L-tyrosine, and their combination significantly increased DNA damage as measured by the increase in percentage (%) of DNA in the comet tail (Fig. 4f). In contrast, cis-RSV treatment did not cause an increase in % of DNA in the comet tail, and attenuated the increase in DNA damage caused by Tyr (Fig. 4f).”*

2. I am not convinced by the role of PARP1, can the authors provide increased serine-PARylation of target proteins in the affected brain regions?

Response: We thank the reviewer for this excellent suggestion. Because we showed decreased levels of HPF1 that is responsible for serine-ADP-ribosylation in the affected brain region of Alzheimer's Disease (AD) patients, we assume that the reviewer would have meant 'decreased' not 'increased' serine-PARylation of target proteins in the affected brain region of AD patients.

Now we show that histone H3 serine-ADP-ribosylation is also *decreased* in the affected brain region (hippocampal brain tissue) of AD patients. This data is incorporated in the revised manuscript as follows. *“Histone poly-ADP-ribosylation factor (HPF1)-dependent serine-ADP-ribosylation is essential for PARP1-dependent DNA repair⁵³ and histone H3 is one of the best characterized targets of HPF1/PARP1-mediated serine-ADP-ribosylation⁵⁴. Because AD brains show accumulation of neuronal DNA damage⁵², we wondered if the levels of HPF1 and histone serine-ADP-ribosylation are affected in AD brains. Our analysis showed reduced levels of HPF1 along with decreased levels of histone H3 serine-ADP-ribosylation in the hippocampal tissues from AD patients (Fig. 5a)”*

3. How does the auto-PARylation of PARP1 that is circadian regulated correlate with DNA damage? PARP1 got its fingers in many pies, how can the authors ascertain these effects to DNA damage and not, for example, to chromatin modification and gene expression?

Response: We agree with reviewer on this point. Because auto-PARylation of PARP1 would also modulate chromatin modification and gene expression, we have now modified our description to include chromatin modification and gene expression also as potential players in the regulation of DNA damage response as follows in the revised draft “*However, PARP1 modulates chromatin modification and gene expression as well. Hence, future studies are required to determine if these functions of PARP1 would also contribute to trans-RSV-mediated neurotoxicity and cis-RSV-mediated neuroprotection.*”

4. There is a confusion between ADP-ribose scars and what the authors describe/observe in their study. The scars that mark sites of DNA damage are mono-ADP-ribose and not poly-ADP-ribose.

Response: We agree with the reviewer on this point. We have now modified our description and eliminated the use of ‘ADP-ribose scars’ from the text.

5. The reduction of HPF1 in hippocampal tissues from AD patients is intriguing. Is it a transcriptional downregulation in response to senescence caused by accumulation of DNA damage? Does the DNA damage accumulate in the HPF1 promoter? The authors need to provide further mechanistic insight to explain this observation.

Response: We thank the reviewer for this excellent suggestion. Our analysis showed that the transcript levels of HPF1 are not affected in AD brains. We have now performed additional experiments to provide a potential molecular basis for the observed reduction of HPF1 levels and serine-ADP-ribosylation in the hippocampal tissues from AD patients. We have incorporated this new data in the revised manuscript as follows. “*Because mRNA levels of HPF1 are not affected in AD brains (Supplementary Fig. 8e) and tyrosine inhibits protein synthesis^{40,41}, we wondered if treatment with tyrosine and trans-RSV would modulate HPF1 at the translational level. Similarly, we also wondered if treatment with DA, BDNF or ISRIB would affect the protein levels of HPF1. We found that while treatment with cis-RSV stimulated serine-ADP-ribosylation, treatment with tyrosine and trans-RSV decreased HPF1 levels along with the inhibition of histone H3 serine-ADP-ribosylation (Fig. 5b-e). Furthermore, treatment with either BDNF or ISRIB increased HPF1 levels along with the induction of histone H3 serine-ADP-ribosylation (Fig. 5f and g). Similarly, DA also stimulated histone H3 serine-ADP-ribosylation (Fig. 5h). Taken together, beyond DNA repair, our data indicate a potential role of serine-ADP-ribosylation in cognition and memory formation and provide a potential molecular basis for the observed reduction of HPF1 levels and serine-ADP-ribosylation in the hippocampal tissues from AD patients.*”

We would like to highlight that H3 serine 10 shows increased phosphorylation (pH3) levels in the affected brain regions of AD patients (PMID: 12677454, PMID: 22760556) and BDNF inhibits the phosphorylation of H3 (PMID: 23741412). These observations suggest that H3 phosphorylation and serine-ADP-ribosylation are opposingly regulated by BDNF. Because

BDNF is depleted (PMID: 30634650) along with concomitant increase in tyrosine levels in AD brains, our data indicate that HPFI might be modulated at the translational level in neurons by the antagonistic effects of tyrosine (*i.e.*; inhibition of the translation of mRNA^{HPF1}) and BDNF (*i.e.*; activation of the translation of mRNA^{HPF1}). Taken together, our new data provide a potential molecular basis for the observed reduction of HPF1 levels and serine-ADP-ribosylation in the hippocampal tissues from AD patients.

6. Why does cis-RSV not affect the recruitment of XRCC1 despite its crucial role in PARP1 mediated response to oxidative damage (Suppl Fig 8e)?

Response: We thank the reviewer for this critical question. Although we did *not* intend to conclude that XRCC1 is not required, we realized that our data from previous figure (Suppl Fig 8e) might have lead to unintended conclusions as highlighted by the reviewer. Therefore, we have now removed data regarding XRCC1 from the figure, while we explore this in future studies.

Reviewer #2 (Remarks to the Author):

Jhanji et al., assemble evidence from the literature that an increase in systemic Tyrosine is associated with cognitive decline, aging, and neurological disorders. They link this observation with findings that Tyrosine mediates protein levels of TyrRS. This effect can be mimicked by Resveratrol (RSV), which has been previously shown to bind to TyrRS and stabilize either TyrRS' Tyr-bound or Tyr-free conformation, dependent on whether RSV is cis- or in trans-configuration. cis-RSV, acting through TyrRS, has positive effects on neuron survival, even under cell stress, while trans-RSV is toxic for neurons at higher concentrations. Among the cell survival promoting effects of cis-RSV/TyrRS, is DNA damage repair through control of PARP1 auto-PARylation and rescue of trapped PARP1.

Overall, it is a beautiful manuscript of high relevance due to the ease with which the development of therapeutics and small molecules can be seen to evolve from this study. The presentation is clear, and the data is convincing. I strongly recommend its publication.

The following points could be addressed to aid with the logical flow of the manuscript. Most comments can be resolved without new experiments and are to aid discussion:

While the regulation of TyrRS protein levels by Tyrosine is supported by literature, the mechanism through which this happens is not clear. Canonically, TyrRS and other tRNA synthetases are viewed as stable proteins with high expressions levels and long turnover times. This might very well be different in neurons and the authors show convincing evidence for TyrRS and PheRS being dynamically regulated. The time scale of TyrRS reduction is fast - 4 h of treatment with neurotoxic agents is sufficient to induce a loss of total TyrRS as shown in S4. It would be interesting to know the underlying mechanism, for example through treatment with proteasome and autophagy inhibitors or inhibitors of translation, such as cycloheximide. TyrRS could also be secreted. If neurons are too sensitive for treatment with sublethal concentrations of

these inhibitors, ubiquitination of TyrRS might be shown by western blot or available mass spectrometry data. Given that the reduction of TyrRS protein is both central to the narrative, quite convincing from the author's data, and counter-intuitive, it would be great to have insights into the molecular regulation behind it, and I would like to see it at least discussed.

Response: We thank the reviewer for these critical comments and excellent suggestions. To gain insights into the molecular regulation behind tyrosine-mediated depletion of TyrRS (YARS1), we have performed additional experiments. Based on this reviewer's comment, we have discussed these results in the manuscript as follows "*Recently published mass spectrometry data showed that TyrRS is heavily ubiquitinated in the cell under stress conditions⁴⁴. Moreover, deubiquitination of TyrRS (along with other aaRSs) facilitates the recovery of translation during the recovery phase⁴⁴. To determine if tyrosine exploits the proteosomal or lysosomal pathway (autophagy) for TyrRS degradation along with its ability to inhibit protein synthesis^{40,41}, we performed additional experiments using inhibitors of autophagy (bafilomycin), proteasome (MG132), and protein synthesis (cycloheximide, CHX). Treatment with both bafilomycin and MG132 increased the levels of TyrRS, suggesting that both the proteasome and lysosome are involved in the constitutive degradation of TyrRS (Supplementary Fig. 3f and g). However, cycloheximide decreased TyrRS levels (Supplementary Fig. 3h), suggesting that sustained de novo synthesis is required to maintain the homeostatic levels of TyrRS under normal conditions whereas deubiquitination may stabilize TyrRS under stress conditions to facilitate the recovery of translation⁴⁴.*"

In line with the above question, is the protein synthesis inhibitory effect of Tyrosine due to the reduction of TyrRS? It is again counter-intuitive why elevated levels of an amino acid should be inhibitory for protein synthesis, for which it is a requisite, therefore, it should be spelled out as clearly as possible in the manuscript.

Response: We thank the reviewer for this suggestion. Yes, we think that the protein synthesis inhibitory effect of tyrosine is due to the reduction of TyrRS protein. Based on reviewer's comment, we now spell out this in the revised draft as follows "*Although it is counter-intuitive that elevated levels of tyrosine is inhibitory for TyrRS and protein synthesis in a p-eIF2 α -independent manner^{40,41}, for which it is required, we also noted that tyrosine co-instantaneously activates the assembly of eukaryotic initiation factor 4 F (eIF4F) and phosphorylation of ribosomal protein S6 kinase beta-1 (S6K1)⁴¹.*"

This is an old question in tRNA synthetase research but given the dramatic drop in TyrRS in Figure 2b I do wonder whether protein synthesis would be affected.

Response: We thank the reviewer for this question. Although we mentioned in the manuscript that *trans*-RSV is known to inhibit protein synthesis, to directly answer the reviewer's question, we tested the effect of *trans*-RSV on global translation using puromycin incorporation assay. While *cis*-RSV had no significant effect on global translation, *trans*-RSV decreased global translation. Therefore, a dramatic drop in TyrRS levels can inhibit protein synthesis. This data is now incorporated in the revised draft as follows "*Finally, consistent with the dramatic depletion of TyrRS and increased levels of p-eIF2 α and p-eEF2, treatment with trans-RSV inhibited global protein synthesis as measured by the puromycin incorporation assay (Fig. 3h)*"

Are there any indications that that would be the case, for example by comparing with viability of siTyrRS neurons?

Response: We thank the reviewer for this question. We have mentioned in the manuscript that siTyrRS does not affect HeLa cell viability based on our previous work. Although we did not observe any viability problems with siTyrRS in neurons using MTT assay, we do report the induction of neurite degeneration due to siTyrRS. We have now made this point explicit in the revised draft as follows. “*Albeit an essential protein, we previously showed that ~75% knockdown of TyrRS using siRNA (siRNA^{TyrRS}) does not affect cell viability²⁸. However, siRNA^{TyrRS} in cortical neurons (~50% knockdown) resulted in robust neurite degeneration (Fig. 1f and Supplementary Fig. 3e), indicating a critical role of TyrRS in maintaining neurite stability.*”

“*Although TyrRS knockdown (Supplementary Fig. 4d) did not significantly affect the viability, it blunted the neuroprotective effects of cis-RSV and did not diminish the toxicity of trans-RSV (50 mM) upon NMDA treatment (Fig. 6c).*”

What are the reasons for all cited clinical studies focusing on trans-RSV over cis-RSV or a racemate? The authors suggest that beneficial results from these studies are based on small amounts of trans-RSV converting into cis-RSV but are detrimental at high concentrations, where the intracellular conversion to cis-sulfate metabolites might be saturated. Elaborating on the reasons for trans-RSV being preferentially studied previously will help put the new results presented here into context.

Response: We thank the reviewer for this excellent suggestion. To help put the new results presented here into context, we have now elaborated on the major reasons in the revised draft in the introduction as follows. “*Calorie restriction (CR) promotes genomic stability through the induction of base excision repair (BER) and reversal of its age-related decline¹⁵ along with extension of lifespan and protection against age-associated neurocognitive and metabolic disorders including cardiovascular diseases (CVD). Although most recent metabolic analyses showed that CR decreases serum tyrosine levels^{16,17}, whether tyrosine has any causal effects in aging and age-associated disorders and their reversal during CR is not yet known. Most importantly, the natural molecule resveratrol (RSV) was shown to evoke CR-like health benefits in humans¹⁸, suggesting that RSV may act as a potential ‘CR mimetic’¹⁸. Intriguingly, clinical studies using the trans-isomer of RSV (trans-RSV) brought out conflicting outcomes, in which lower doses of trans-RSV produced encouraging results but higher doses exacerbated the diseases¹⁹. For example, low-dose trans-RSV showed CR-like benefits in obese males¹⁸, cognitive benefits in AD patients²⁰ and in postmenopausal women²¹, and protection against heart failure²². However, higher doses (≥ 200 mg/day) of trans-RSV resulted in brain volume loss in AD patients²³, worsened memory performance in schizophrenia²⁴, and increased CVD risk²⁵. Despite decades of research, the molecular basis of these controversial effects of trans-RSV (low dose CR-like beneficial effects^{18,20-22} versus high dose detrimental effects²³⁻²⁵) remains unknown¹⁹. Although RSV exists as a mixture of two isomers (cis-RSV and trans-RSV), recent studies demonstrated that the sulfate metabolites of trans-RSV that provide an intracellular pool²⁶ mainly generate cis-RSV²⁷. We previously showed that cis- and trans-RSV mimic tyrosine in binding to the active site*”

of TyrRS²⁸. However, our recent analysis of the x-ray crystal structures of TyrRS with and without cis-RSV showed that the binding of cis-RSV in the active site of TyrRS mimics its 'tyrosine-free' conformation¹⁹. Since CR decreases serum tyrosine levels^{16,17}, we proposed that binding of cis-RSV may enable the moonlighting functions of TyrRS even in the presence of tyrosine and hence, cis-RSV may act as a 'CR mimetic'^{19,28}.

We previously showed that tyrosine inhibits the moonlighting functions of TyrRS that activate the auto-poly-ADP-ribosyl(PAR)ylation of poly-ADP-ribose polymerase 1 (PARP1) and associated stress signaling²⁸. Consistently, auto-PARylation of PARP1 is circadian-regulated in a feeding-dependent manner, in which feeding that increases tyrosine levels inhibits auto-PARylation²⁹. These observations suggested that the moonlighting functions of TyrRS are normally activated when tyrosine levels are decreased during the nadir/trough of circadian rhythm. However, we also showed that lower concentration trans-RSV adapts its cis conformation (cis-RSV) to activate TyrRS-dependent auto-PARylation of PARP1²⁸. Although auto-PARylation of PARP1 is essential for BER³⁰ and sleep activates neuronal DNA repair⁷ whether cis-RSV would enable the moonlighting function of TyrRS in DNA repair is not yet known. Moreover, the apparent K_i value of trans-RSV-mediated inhibition of tyrosine activation by TyrRS in an ATP-PPi exchange assay ($\text{Tyr} + \text{ATP} \rightarrow \text{Tyr-AMP} + \text{PPi}$) is $\sim 25 \mu\text{M}$ ²⁸, suggesting that trans-RSV may retain its trans conformation (mimicking 'tyrosine-like' conformation) at the active site of TyrRS at higher concentrations ($\geq 25 \mu\text{M}$)^{19,28}. However, whether trans-RSV inhibits the moonlighting functions of TyrRS in DNA repair, especially the activation of auto-PARylation of PARP1^{19,28} is not yet known. Most importantly, clinical studies using 5 and 1000 mg of trans-RSV (which is >99% trans-RSV) reported peak plasma concentrations of 0.6 and 137 μM of RSV respectively³¹, and other clinical studies using 1000 mg/day of trans-RSV for 29 days reported an accumulation of 50-640 μM of trans-RSV in human tissues²⁶, suggesting that this treatment regimen could achieve high dose ($\geq 25 \mu\text{M}$) trans-RSV-mediated effects in humans."

The link to eEF2 and eIF2 could be explained better, preferably with a scheme –

Response: We thank the reviewer for this excellent suggestion. We have now included a scheme to explain the link to eEF2 and eIF2 and protein synthesis and provide an explanation in the revised draft as follows. "Protein synthesis is mainly regulated at the initiation and elongation steps. Ser51 phosphorylation of eukaryotic initiation factor 2 alpha (p-eIF2 α) by multiple kinases and Thr56 phosphorylation of eukaryotic elongation factor 2 (p-eEF2) by eEF2 kinase (eEF2K) inhibit protein synthesis at the initiation and elongation steps, respectively. The protein kinase mammalian target of rapamycin (mTOR) inhibits eEF2K to activate protein synthesis (Supplementary Fig. 4a)."

I assume dopamine does not chemically dephosphorylate eEF2 itself but instead induces its dephosphorylation through a pathway?

Response: We agree with the reviewer that dopamine does not chemically dephosphorylate eEF2 itself and we also apologize for not making it clear. We have now described the process in the revised draft as follows. "Interestingly, DA is known to activate eEF2⁴⁵, potentially by activating mTOR, resulting in the stimulation of neuronal protein synthesis and memory formation⁴⁶."

Was eEF2K chosen as the target of pharmaceutical inhibition because an inhibitor was readily available or due to its specific function in the eEF2 complex?

Response: We apologize for not making it explicit. eEF2K was chosen due to its specific function in the eEF2 complex. We have now made it clear in the revised draft as follows. “*Because eEF2 regulates the elongation phase of protein synthesis and eEF2K is the only known kinase that inhibits eEF2 (Supplementary Fig. 4a), we wondered if direct pharmacological activation of eEF2K using nelfinavir⁴⁷ would deplete TyrRS.*”

TyrRS is among the ATF4 target genes that are specifically upregulated in ISR while general translation is globally reduced (Han, ... Kaufman, 2013). It is surprising that TyrRS levels are affected positively by the suppression of ISR – are there indications that the regulation of gene expression in ISR is different in neurons or is it a concentration dependent effect as the authors do see a repression of TyrRS upon higher doses of ISRIB?

Response: We thank the reviewer for this excellent suggestion and agree with the observation that TyrRS is among the ATF4 target genes that are specifically upregulated in ISR. We have now highlighted that TyrRS is a target of ATF4 in the revised draft as follows “*Interestingly, TyrRS (but not PheRSb) is among the activating transcription factor 4 (ATF4) target genes that are specifically upregulated in integrated stress response (ISR)⁵⁰. Because we have no indication that the regulation of gene expression in ISR is different in neurons, and ISRIB inhibits ATF4 target gene expression⁴⁹, it is highly likely that the concentration dependent decrease in TyrRS level is due to the repression of TyrRS upon higher doses of ISRIB.*”

Tyrosine induced DNA damage is especially interesting in neurons as they accumulate over time due to the long life of neuronal cells. Together with the findings on removal of ADP-ribose scars, I’m wondering to which extent RSV is able to reverse DNA damage or if it is mostly protective towards the further accumulation of DNA damage.

Response: We thank the reviewer for this interesting suggestion. Our new data shows a substantial reduction of γ -H2AX staining in neurons even after 12 hr of pre-treatment with tyrosine that caused substantial amount of DNA damage. This data suggests that beyond preventing, *cis*-RSV may potentially also reverse existing DNA damage. We have incorporated this new data in the revised draft as follows “*Interestingly, cis-RSV reduced γ -H2AX levels in neuronal cultures even after 12 hr of tyrosine pre-treatment that had already caused substantial amount of DNA damage (Supplementary Fig. 8a), suggesting that beyond prevention, cis-RSV may reverse existing neuronal DNA damage as well.*”

Minor issues/typos:

It should be mentioned that the official IUPAC name of TyrRS is YARS1.

Response: Thank you, we have now indicated the IUPAC name of TyrRS is YARS1 in the maintext.

It could be stated more clearly in the figure legends whether n-numbers refer only to one panel or to the whole figure (for example 1h, mean \pm SEM, n=3 for all figures or only figure 1h?).

Response: Thank you, we have now indicated more clearly that n-numbers refer to the whole figure.

Some material and method sections are justified, others are not.

Response: Thank you, we have now justified everything in the material and method sections.

Figure S1F has a typo: Brain instead of Brian

Response: Thank you, we have now corrected the typo.

Use of publicly available proteomics data... has spaces between citation and text, which are missing in the rest of the document.

Response: Thank you, we have now removed the space between citation and text.

S3f legend lacks a space between “(rapa, 100nM)”

Response: Thank you, we have now incorporated the space between “(rapa, 100nM)”.

Cell viability assays: surplus space between “16 hr.” and “Cultured rat cortical”

Response: Thank you, we have now removed surplus space between “16hr.” and “Cultured rat cortical”.

Reviewer #3 (Remarks to the Author):

In this manuscript, Jhanji et al. shows that increased tyrosine levels cause neurotoxicity and DNA damage in neurons by modulating Tyrosyl-tRNA synthetase levels. Next, the authors show that trans-Resveratrol induces a neurotoxic high-tyrosine like state with increased DNA damage whereas cis-Resveratrol induces protective low-tyrosine like state, both mediated through their binding to Tyrosyl-tRNA synthetase. Finally, they demonstrate that the increased DNA damage by trans-Resveratrol is mediated by trapped PARP1 on broken ends of DNA. Overall, this is a highly relevant report highlighting the dose and trans vs cis isomer dependent effect of a widely popular anti-aging supplement and I recommend publication after addressing the comments noted below.

1) The authors hypothesize lower Tyr may switch the function of a fraction of TyrRS from protein synthesis to DNA repair. Does TyrRS in Tyr-free state translocate to the nucleus if it acts as a DNA repair factor? Can the authors show evidence for this at low Tyr concentrations?

Response: We thank the reviewer for this excellent suggestion. We have now performed additional experiments. Our data indicate that low Tyr concentration is sufficient to translocate TyrRS to the nucleus. We have now incorporated this data in the revised draft as follows. *“Conversely, reducing tyrosine levels in the culture medium increased TyrRS levels in the nucleus and neurites (Fig. 1c and d and Supplementary Fig. 3a).”*

2) The hypothesis for the neuroprotective effects of cis-RSV is that it enhances DNA repair through TyrRS. The authors show that treatment with cis-RSV enhances the cell survival after Etoposide treatment (Fig S7a). Can the authors show that this is indeed mediated by enhanced DNA repair by measuring γ H2AX signal after cis-RSV+Etoposide treatment?

Response: We thank the reviewer for this suggestion. We have done additional experiments to show that the cell survival is indeed mediated by enhanced DNA repair by measuring γ H2AX signal after cis-RSV+Etoposide treatment. We have now incorporated this new data in the revised draft as follows. *“We further confirmed that cis-RSV-mediated rescue of ETO-mediated DNA damage is reflected by decreased levels of γ -H2AX (Supplementary Fig. 9a).”*

3) Similar to comment 1, does cis-RSV increase translocation of TyrRS into nucleus to enhance DNA repair.

Response: We thank the reviewer for this suggestion. Yes, cis-RSV increases the translocation of TyrRS into the nucleus as we have now measured the nuclear levels of TyrRS after cis-RSV treatment and incorporated this data in the revised draft as follows. *“Consistent with the results in clinical trials, a low dose of trans-RSV (10 μ M) increased TyrRS and the high dose (50 μ M) decreased it, whereas cis-RSV (10-50 μ M) increased TyrRS protein levels both in the nucleus and neurites (Fig. 3a and b and Supplementary Fig. 5a and b) mimicking the effects of reduced levels of tyrosine that increase TyrRS in the nucleus and neurites (Fig. 1c).”*

“Although cis-RSV-mediated auto-PARylation of PARP1 resulted in its removal from the chromatin, unexpectedly, we found that cis-RSV activated the deADP-ribosylation of the chromatin fraction along with higher levels of TyrRS (Fig. 7a and Supplementary Fig. 10b).”

4) The authors show that PARP1 leads to DNA damage and neurotoxicity through inhibition of DNA repair. However, the model figure in Figure 5e tends to indicate that trans-RSV actively induces DNA damage. Please clarify the model figure or clarify in text.

Response: We thank the reviewer for this critical observation and suggestion. We have now clarified it in the model figure showing that ‘trapped’ PARP1 induced DNA damage and neurotoxicity.

Minor:

1) Indicate γ H2AX is green in figure 3a. It’s confusing that cis-RSC is also colored green. Perhaps change the color for γ H2AX signal.

Response: Thank you, we have now changed the coloring pattern.

2) How was 8-oxo-dG level measured, please indicate in figure legend.

Response: Thank you, we have now indicated 8-oxo-dG level measured using immunofluorescence (IF) in the figure legend.

Reviewers' Comments:

Reviewer #2:

Remarks to the Author:

The authors have gone above and beyond to address my comments. I strongly recommend its publication.

Reviewer #3:

Remarks to the Author:

The authors have now satisfactorily addressed my comments. However, there appears to be errors in the figure legends and figure citations. Please review the manuscript for such errors before publication in Nature communications.

- 1) Check figure legends for Fig 1c and d. Description for figure 1d seems to be marked as 1c.
- 2) In the response to comment 2, the authors refer to Supplementary Fig 9a. I believe they meant Supplementary Fig 8a. Please check in the manuscript if the figure is cited properly.

Reviewer #4:

Remarks to the Author:

The paper by Jhanji et al. explores the roles of tyrosine in neurodegeneration with links to ageing. In this review, I am considering solely the response of the authors to the comments/suggestions of Reviewer 1.

The authors have responded to all the requests by this reviewer. A few comments on two points:

1. Reviewer 1 asked the authors to perform comet assays to discriminate between SSBs to DSBs. I think the reviewer should have mentioned the need to perform a neutral comet assay rather than alkaline. The authors performed an alkaline assay that doesn't discriminate between such differences. In fact, while the neutral comet assay is mostly used to detect double-stranded DNA breaks, the alkaline comet assay is more sensitive for smaller amounts of DNA damage, including a variety of lesions ranging from SSBs and DSBs to alkali-labile sites, DNA-DNA or DNA-protein cross-linking. The best way to discriminate in this instance between these two lesions would have been co-staining with 53BP1. A co-stain with a neuronal marker and/or a G1 cell cycle marker would have made this panel even better. That being said, I believe this is a minor point and is clear that trans-RSV increases the DNA-damage load. I would suggest the authors to separate the two channels in figure 4a and show the DAPI and gH2AX in separate images with traced nuclear shape in the gH2AX-nonDAPI image. Especially in the print version, this will increase the visibility of the number of foci for the reader (apparent on the magnified pdf electronic version but very hard to see in print).

2. The model presented by the authors is one in which PARP1 is trapped. Would have been nice for the authors to use a PARP1i (i.e. olaparib) to potentiate this point. Olaparib traps PARP1 to chromatin/DNA and in theory should mimic this effect.

We thank the reviewers for their great suggestions which have significantly improved the paper and hope we have satisfactorily addressed the remaining points raised by the reviewers.

Reviewer's comments are in blue italics below and our responses are in black.

Reviewer #2 (Remarks to the Author): The authors have gone above and beyond to address my comments. I strongly recommend its publication.

Response: Thank you.

Reviewer #3 (Remarks to the Author): The authors have now satisfactorily addressed my comments.

Response: Thank you.

However, there appears to be errors in the figure legends and figure citations. Please review the manuscript for such errors before publication in Nature communications. 1) Check figure legends for Fig 1c and d. Description for figure 1d seems to be marked as 1c.

Response: We apologize for overseeing the errors in Figure legends. We have now corrected them.

2) In the response to comment 2, the authors refer to Supplementary Fig 9a. I believe they meant Supplementary Fig 8a. Please check in the manuscript if the figure is cited properly.

Response: Thank you. We have now corrected them in the manuscript.

Reviewer #4 (Remarks to the Author): The paper by Jhanji et al. explores the roles of tyrosine in neurodegeneration with links to ageing. In this review, I am considering solely the response of the authors to the comments/suggestions of Reviewer 1. The authors have responded to all the requests by this reviewer.

Response: Thank you.

A few comments on two points: 1. Reviewer 1 asked the authors to perform comet assays to discriminate between SSBs to DSBs. I think the reviewer should have mentioned the need to perform a neutral comet assay rather than alkaline. The authors performed an alkaline assay that doesn't discriminate between such differences. In fact, while the neutral comet assay is mostly used to detect double-stranded DNA breaks, the alkaline comet assay is more sensitive for smaller amounts of DNA damage, including a variety of lesions ranging from SSBs and DSBs to alkali-labile sites, DNA-DNA or DNA-protein cross-linking. The best way to discriminate in this instance between these two lesions would have been co-staining with 53BP1. A co-stain with a neuronal marker and/or a G1 cell cycle marker would have made this pannel even better. That being said, I believe this is a minor point and is clear that trans-RSV increases the DNA-damage

load. I would suggest the authors to separate the two channels in figure 4a and show the DAPI and gH2AX in separate images with traced nuclear shape in the gH2AX-nonDAPI image. Especially in the print version, this will increase the visibility of the number of foci for the reader (apparent on the magnified pdf electronic version but very hard to see in print).

Response: We agree and thank the reviewer for this great suggestion. We have now separated the two channels in figure 4a (and in figure 8c) showing the DAPI and γ H2AX in separate images with traced nuclear shape in the γ H2AX-nonDAPI image.

2. The model presented by the authors is one in which PARP1 is trapped. Would have been nice for the authors to use a PARP1i (i.e. olaparib) to potentiate this point. Olaparib traps PARP1 to chromatin/DNA and in theory should mimic this effect.

Response: We thank the reviewer for this excellent suggestion. We have now show that olaparib and other PARP inhibitors that trap PARP1 to chromatin/DNA which in theory mimic the effect of *trans*-resveratrol-mediated neurotoxicity in HR-deficient post-mitotic neurons. We have now incorporated these data in the revised draft as follows. “**PARP inhibitors are neurotoxic in HR-deficient post-mitotic neurons.** Post-mitotic neurons are homologous recombination (HR)-deficient⁵⁷ and utilize non-homologous end-joining (NHEJ) for DNA repair. Interestingly, H3 serine-ADP-ribosylation facilitates NHEJ⁵⁸ and Ku-dependent DNA repair is inhibited in AD brains⁵¹. While depletion of PARP1 increases HR⁵⁶, PARP inhibitors drive toxic NHEJ in HR-deficient cells in an ataxia-telangiectasia mutated (ATM)-dependent manner⁵⁹. Because we previously showed that TyrRS activates ATM through acetylation²⁸ and cis-and trans-RSV have opposite effects on TyrRS levels (**Fig. 3a and b**) and the auto-PARylation of PARP1 (**Fig. 6e and f**), we tested the effect of well-known PARP inhibitors on cis-and trans-RSV-mediated effects on rat cortical neurons. While treatment with PARP1-specific inhibitor AG-14361 (AG) did not affect cis-RSV-mediated neuroprotective effects, treatment with olaparib (Ola) that inhibits both PARP1 and 2 mitigated the neuroprotective effects of cis-RSV (**Fig. 8a**) and did not affect trans-RSV-mediated neurotoxicity (**Fig. 8b**). Because siRNA^{PARP1} mitigated the effect of trans-RSV (**Fig. 7d**), taken together, these results indicate a critical neuroprotective role of PARP2, which may be utilized by cis-RSV in the absence of PARP1. Moreover, we found that PARP inhibitors themselves decreased neuronal survival (**Fig. 8a and b**), induced neuronal DNA damage (**Fig. 8c**) and neurite degeneration (**Fig. 8d**), suggesting that PARP inhibitors trigger cell death in post-mitotic HR-deficient neurons through the inhibition of H3 serine-ADP-ribosylation-dependent NHEJ⁵⁸”

Reviewers' Comments:

Reviewer #4:

Remarks to the Author:

The authors addressed all my suggestions.